# Deglaciation-enhanced mantle $CO_2$ fluxes at Yellowstone imply positive climate feedback

Fiona Clerc [1,2] ✉, Mark D. Behn [3] & Brent M. Minchew[4]

Mantle melt generation in response to glacial unloading has been linked to enhanced magmatic volatile release in Iceland and global eruptive records. It is unclear whether this process is important in systems lacking evidence of enhanced eruptions. The deglaciation of the Yellowstone ice cap did not observably enhance volcanism, yet Yellowstone emits large volumes of $CO_2$ due to melt crystallization at depth. Here we model mantle melting and $CO_2$ release during the deglaciation of Yellowstone (using Iceland as a benchmark). We find mantle melting is enhanced 19-fold during deglaciation, generating an additional 250–620 km$^3$. These melts segregate an additional 18–79 Gt of $CO_2$ from the mantle, representing a ~3–15% increase in the global volcanic $CO_2$ flux (if degassed immediately). We suggest deglaciation-enhanced mantle melting is important in continental settings with partially molten mantle – including Greenland and West Antarctica – potentially implying positive feedbacks between deglaciation and climate warming.

The coupling between the solid Earth, cryosphere, and climate is key to understanding past changes in these systems and their future evolution. In particular, as an ice mass retreats and unloads the Earth's surface, the underlying mantle rebounds and undergoes a reduction in pressure. If the mantle is above the solidus, this decompression generates additional melting relative to the background melting rate. Enhanced mantle melting can result in increased volcanic activity[1,2], which in turn may incite the release of aerosols into the atmosphere, the acceleration of glacier flow by geothermal heating, and outburst flooding from glacial lakes. The rapid flow of the Northeast Greenland Ice Stream has been attributed to elevated geothermal heat fluxes (GHF) due to volcanism[3] or the passage of the Iceland plume, perhaps influencing the mass of the Greenland Ice Sheet over glacial-interglacial cycles[4]. Beneath the West Antarctic Ice Sheet (WAIS), ice flow could be enhanced by elevated GHF from subglacial volcanism[5,6] or a mantle plume[7]. Understanding whether deglaciation enhances continental and/or hotspot magmatism has implications for the retreat of the Greenland and West Antarctic Ice Sheets.

Increased mantle melting also enhances the extraction of $CO_2$ from the mantle. If released to the surface, the additional magmatic $CO_2$ can impact the Earth's climate. During the last deglaciation, subaerial volcanoes are thought to have erupted up to 1000–5000 Gt of additional $CO_2$ (refs. [8,9]). Changes in sea-level associated with glacial-interglacial cycles may also enhance $CO_2$ emissions from mid-ocean ridge volcanoes[10]. However, little work has focused on the enhancement of diffuse subaerial $CO_2$ emissions from hydrothermal systems and dormant volcanoes, despite their large present-day $CO_2$ flux of 170 Mt/yr, representing roughly half of the modern global volcanic $CO_2$ flux[11].

The link between deglaciation and enhanced mantle melting is most strongly established in Iceland[1,2,12], where increases in eruptive volumes coincide with the most rapid stage of the Late Weischelian deglaciation of the Iceland ice sheet from 11–10 ka (BP). While shallower crustal processes may also modulate the magmatic response to deglaciation, the importance of enhanced mantle melting is evidenced by the magnitude of deglacial eruptive rates and the coeval depletion of incompatible trace elements, first modelled by Jull and McKenzie[1] (hereafter JM96).

By comparison, deglaciation-enhanced melting in continental mantle has not been quantified (with the exception of global ice mass loss scalings[8]), and observations of enhanced volcanism during deglaciation in intraplate settings are primarily attributed to the

[1]Previously at: MIT-WHOI Joint Program in Oceanography/Applied Ocean Science & Engineering, Cambridge, MA, USA. [2]Lamont-Doherty Earth Observatory, Columbia University, Palisades, NY, USA. [3]Department of Earth and Environmental Sciences, Boston College, Chestnut Hill, MA, USA. [4]Department of Earth, Atmospheric and Planetary Sciences, Massachusetts Institute of Technology, Cambridge, MA, USA. ✉e-mail: fclerc@ldeo.columbia.edu

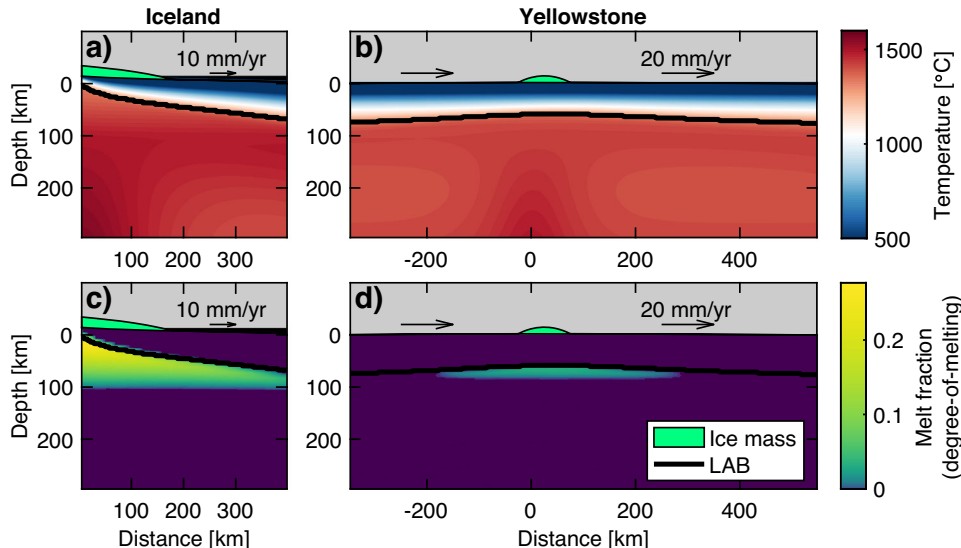

**Fig. 1 | Background mantle temperatures and melt fractions (i.e., "degree-of-melting"), prior to unloading.** Temperatures beneath (**a**) Iceland and (**b**) Yellowstone are plotted in red-blue. The thick black line is the lithosphere-asthenosphere boundary (LAB), at T = 1300 °C. The green parabola represents the ice volume at its maximum (10-fold vertical exaggeration). Black arrows indicate imposed plate motions. Melt fractions in blue-green plotted for (**c**) Iceland and (**d**) Yellowstone.

triggering of crustal magma chambers[13,14]. For example, Yellowstone is magmatically active and has experienced rapid deglaciation. During the Pinedale (22–13 ka) and Bull Lake (140–150 ka) glaciations, ice caps covered the Yellowstone caldera and beyond, extending ~100 km in radius[15]. While the Pinedale deglaciation occurred during a period of volcanic quiescence, the Bull Lake deglaciation occurred during the most recent eruptive episode in Yellowstone, the Central Plateau Member rhyolites (170–70 ka). Geological evidence suggests many of these eruptions are syn-glacial[16,17], although limited age constraints hinder interpretations of correspondence between glaciation and volcanism. The Central Plateau Member rhyolites were erupted from a large upper crustal sill, maintained by an extensive deeper magmatic system potentially fed by a mantle plume[18]. During the deglaciation interval there is no evidence that eruptive rates were heightened, nor that the magmatic system was otherwise altered, relative to background rates/trends. However, Yellowstone's present-day magmatic $CO_2$ flux (~3% of the modern global flux[19]) is released not by eruptions, but by the crystallization of magmas at depth[20]. Thus, it remains unclear whether mantle melting rates and associated volatile fluxes are significantly enhanced under thicker continental lithosphere, particularly as glacially induced pressures are attenuated with depth[1], and by extension whether the singularly strong response of Iceland is related to the unique juxtaposition of the Icelandic mantle plume and the Mid-Atlantic ridge.

To gain insight into the potential for enhanced magmatism and $CO_2$ release during the last deglaciation of Yellowstone, we model the influence of ice retreat on mantle melting in both Iceland and Yellowstone. The Iceland model is compared against eruption rates, geochemical observations, and previous theoretical studies associated with the deglaciation of Iceland. Iceland serves as an ideal benchmark case because its location, co-located with the mid-Atlantic ridge axis, promotes rapid extraction of melts to the surface. Second, we explore the conditions conducive to the enhancement of melting beneath thick continental lithosphere such as in Yellowstone (where melt extract to the surface may be inhibited), and examine implications for such magmatism beneath modern ice sheets.

We use the mantle convection code ASPECT[21,22] to simulate changes in pressure and melt production due to glacial unloading (see Methods). The models are first run to steady-state to resemble present-day "background" behavior (Fig. 1) and are then loaded/unloaded using the reconstructed ice load for each system. The models are unloaded by decreasing the ice sheet radius at a constant rate over a prescribed deglaciation interval (1000 years for Iceland, 2000 years for Yellowstone), simulating the retreat of the ice margin. The mantle melt production rate is the rate of melt fraction change integrated spatially (here melt fraction is defined as "degree-of-melting" and is a thermodynamic property of the solid mantle). Because of the co-located ridge-plume environment, we report melt production rates in 2–D for Iceland, with a suggested 3–D scaling. For Yellowstone, we report radially integrated rates in 3–D, which are verified against a full 3–D model. We also calculate trace element concentrations and estimate the flux of $CO_2$ segregated from the mantle by melts and the flux of $CO_2$ exsolved to the surface. Finally, we estimate the heat released by the emplacement of additional melts.

## Results and discussion
### Deglaciation melting in Iceland
We first model mantle melt production rates underneath Iceland (Fig. 2; "primary run") and benchmark our approach against JM96 (Supplemental Information). Prior to unloading, the mantle flow field is a combination of passive corner flow from plate spreading and dynamic flow from the thermally buoyant plume (red arrows in Fig. 2a). The integrated background melting rate is $0.9 \times 10^{-3}$ km²/yr, assuming only melts generated within 30 km of the ridge axis reach the surface[23,24]. Integrating this 2-D rate over a 100 km zone along the ridge axis (which is narrower than the width of the plume source[25]) yields a 3-D volumetric flux rate of ~0.09 km³/yr (orange line in Fig. 3b; see Methods).

As the mass of the ice sheet is unloaded, the underlying mantle rebounds (Fig. 2c, red arrows), inducing large rates of decompression (Fig. 2c, teal). The background flow is still present but is overshadowed by the much greater (>0.3 m/yr) glacial isostatic adjustment. Due to the thin lithosphere, the mantle response is localized, roughly confined within the margin of the retreating ice sheet. The large rates of decompression greatly enhance melt production rates (Fig. 2d) throughout the ridge melting triangle. When spatially integrated throughout the entire domain, the melt production rate increases by an "enhancement factor" of ~18 during the deglaciation interval, producing 0.017 km²/yr (~1.7 km³/yr) of melt (Fig. 3b, black line). JM96

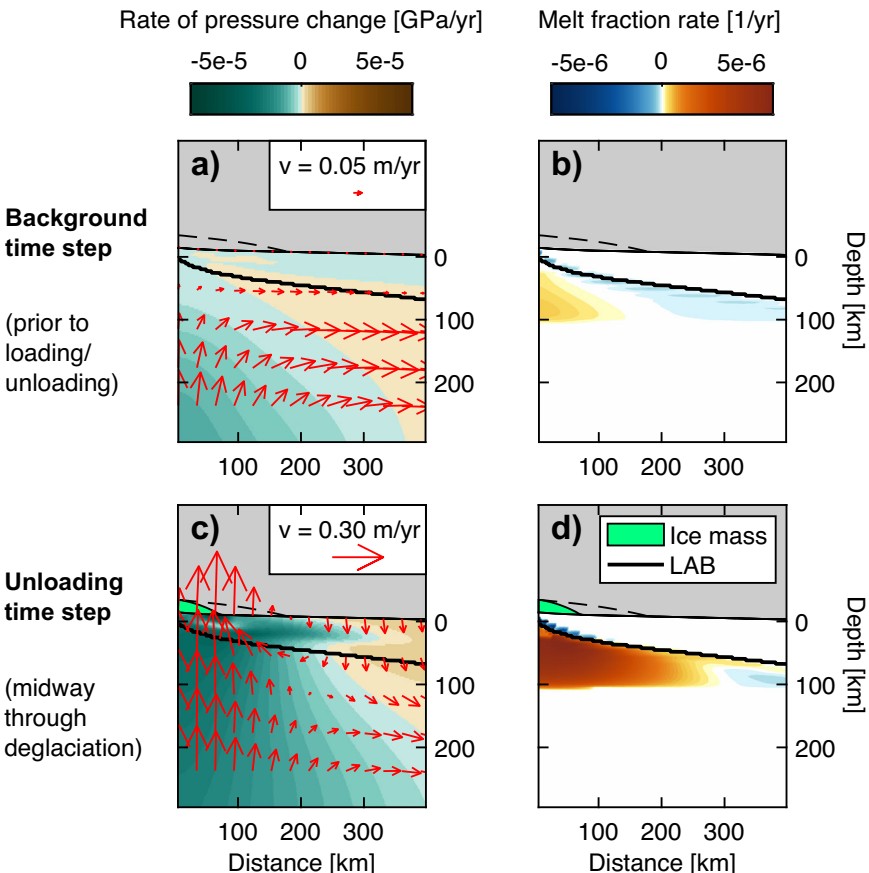

**Fig. 2 | Modeled melt production due to deglaciation of Iceland ice sheet.** The ice sheet is represented by the green parabola at a given time step and by the dashed black line at its maximum extent. Rates of pressure change are colored teal-brown (**a**, **c**) and rates of melt fraction change are colored blue-orange (**b**, **d**). Top row shows a model time step prior to any glacial loading/unloading, while bottom row shows a time step 500 years following deglaciation onset. Red arrows show mantle flow (arrow size is scaled to velocity magnitude); the thick black line is the lithosphere-asthenosphere boundary (LAB), at T = 1300 °C.

predict similar increases in melt production during deglaciation using slightly different model assumptions (see Section S1).

Overall, we find that the rates of enhanced melt production depend primarily on the thermal structure and background melt fractions prior to deglaciation, and the total rate and volume of ice removed. We test different styles of ice sheet retreat (Fig. S4), but find that the total melt production by the end of deglaciation scales most closely with the total change in ice sheet volume. Under larger spreading rates or mantle temperatures, melt fractions increase and the zone of enhanced melting broadens in horizontal extent. Yet the relative enhancement in melting is smaller under these more productive conditions (Fig. S5).

We estimate the concentration of $CO_2$ in the melt and the flux of $CO_2$ released to the surface. We calculate the partitioning of $CO_2$ into the melt using a retained melt fraction formulation[26], which can reproduce the magnitude of the observed[2,27] depletion in trace element concentrations due to deglaciation (see Methods and Figure S14a). Our background $CO_2$ fluxes are within the range inferred from helium fluxes[28], subject to uncertainties in scaling our 2−D model to a 3−D flux. During the deglaciation, we calculate that for a mantle $CO_2$ content of 150–600 ppm (see Methods), an additional 0.05–0.44 Gt/km (-5–44 Gt) of $CO_2$ is released over 1 kyr (dash-dotted black line, Fig. 3d), corresponding to a 6-fold increase over the background flux. The release of this additional $CO_2$ is likely not instantaneous, but is slowed by processes such as melt migration[29]. This excess flux is smaller than a prior estimate employing a 1−D melt column[29], which found an extra -165 Gt $CO_2$ was released over the 11 kyrs following deglaciation for a mantle $CO_2$ content of 285 ppm.

Finally, we examine the conditions under which the heat released by the emplacement of the additional melts may reach the surface. The emplacement of our steady-state melt production rate at a depth of 10 km releases 0.007 GW/km (-7 GW) of heat (comparable to a similar calculation[30] of 8 GW). This flux may be transferred conductively to the surface over long time scales, consistent with borehole measurements from outside the rift zone[30]. During the deglaciation, we estimate the emplacement of the additional melt releases 1.2 GW/km (-120 GW) at depth, for a total of $4 \times 10^{19}$ J/km (-$4 \times 10^{21}$ J) over the entire interval. For comparison, the energy required to melt a 100,000 km³ Icelandic ice sheet near its melting point is $30 \times 10^{21}$ J.

**Deglaciation melting in Yellowstone**

We next estimate how deglaciation affects mantle melt production rates associated with the Yellowstone plume. Prior to unloading, the background mantle flow field represents a combination of shearing from the westward motion of the North American plate and upwelling from the plume (Fig. 4a, red arrows). Melts are produced at depths of 90 to 70 km (consistent with ref. 31), over a 300 km wide region (orange colors in Fig. 4b). The background mantle melt production rate of 0.022 km³/yr (0.00018 km²/yr in 2−D) represents the rate of emplacement of basalts (consistent with refs. 20,32), assuming efficient melt extraction.

During the deglaciation, we find that the enhancement of melting beneath Yellowstone is comparable to Iceland (Fig. 3b, c), despite the thickness of the continental lithosphere and the smaller rates of unloading from the Yellowstone ice cap. The upper asthenosphere rises at a rate of 7 cm/yr due to a combination of the background

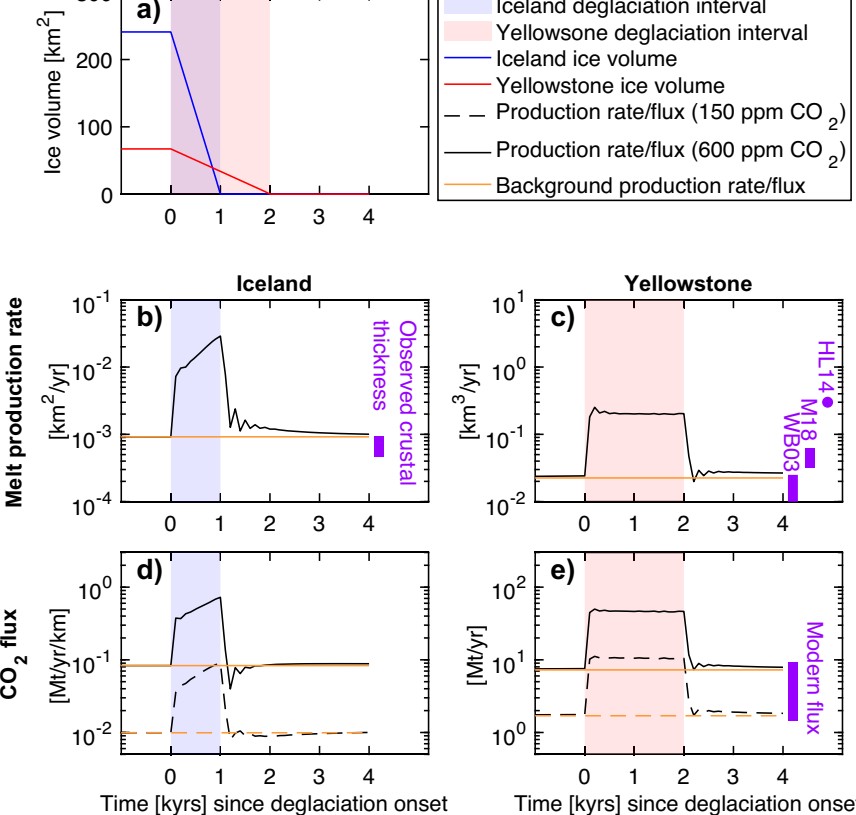

**Fig. 3 | Evolution of melt production rate and $CO_2$ flux during deglaciation. a** Ice volumes used as model forcings for Iceland (blue) and Yellowstone (red) during the deglaciation intervals (shaded). Melt production rates (black lines) for (**b**) Iceland and (**c**) Yellowstone; background rates from time steps prior to loading/unloading are plotted in orange. Note the different scales (2–D for Iceland; 3–D for Yellowstone). refer to Figure S9b for 3–D Yellowstone ice volumes. $CO_2$ fluxes for (**d**) Iceland and (**e**) Yellowstone assuming mantle source $CO_2$ concentrations of 150 and 600 ppm are plotted as dashed and solid lines, respectively. Estimates of modern melt production rates and magmatic $CO_2$ fluxes for Iceland[63,64] and Yellowstone[20,68,69] are denoted by purple bars. "WB03" refers to Werner & Brantley[20], "M18" refers to McMillan et al.[68], "HL14" refers to Hurwitz & Lowenstern[69].

plume/plate flow and isostatic adjustment (Fig. 4c). The zone of positive melt production grows laterally and extends to shallower depths of 60 km (Fig. 4d). The total melt production rate increases to 0.20 km³/yr (0.012 km²/yr) during deglaciation, representing a 9-fold enhancement of melting and an additional 360 km³ of melt over the entire deglaciation (Fig. 3c). Subject to uncertainties associated with the 3–D ice cap geometry (Section S3.5), and with the background rate of melt production (Section S3.1), we estimate this volume of additional melting falls within the range of 250–620 km³. Modelled trace element profiles predict a ~ 30% depletion in light rare Earth elements (LREE) during unloading, relative to background basalt compositions (Fig. S14b).

We also test the response of a transient upper mantle thermal anomaly without a plume tail (Fig. S8) and higher melt production rates (Fig. S7). In the case lacking a plume tail, unloading of the transient upper mantle thermal anomaly yields melt production rates that are almost as high (93%) as the case with a plume tail (Fig. S8). In cases with higher melt production rates, greater volumes of additional melt are generated during deglaciation (see Methods).

The enhancement in melt production implies more carbon is extracted from the mantle and released to the surface as $CO_2$. Extrapolated surface measurements of diffuse outgassing at Yellowstone[19] predict a modern-day $CO_2$ flux of 4–13 Mt/yr (an earlier estimate[20] predicts 11–22 Mt/yr). Carbon and helium isotopes suggest that ~50–70% of the diffuse outgassing flux may be attributed to mantle magmatism[20]. Assuming mantle $CO_2$ concentrations of 150–600 ppm (within the range observed in mantle xenoliths[33]), we obtain background mantle-derived

$CO_2$ fluxes of 1.7–7.3 Mt/yr, in agreement with the above constraints. During unloading, the $CO_2$ flux increases to 11–47 Mt/yr, representing a 6-fold enhancement if released during the deglaciation. Over the entire deglaciation, we estimate the release of an additional 18–79 Gt $CO_2$ to the surface.

The large enhancement in melting may transfer additional heat from the mantle to the crust or surface. Melts derived from the mantle are thought to recharge a large upper crustal sill, imaged seismically at depths of 4–14 km (ref. 18). We estimate the emplacement of the 0.022 km³/yr background melt production rate at a depth of 14 km releases 3.8 GW of heat, comparable to the 4–8 GW extrapolated from chloride fluxes[34]. During deglaciation, the emplacement of the additional melts would impart an additional 31 GW of heat at depth, for a total of $2 \times 10^{21}$ J over the deglaciation interval. The energy required to melt a 20,000 km³ Yellowstone ice cap near its melting point is $6 \times 10^{21}$ J.

### Deglaciation melting in continental settings
Our calculations imply that Yellowstone underwent a sizable enhancement in melting due to deglaciation, slightly smaller than the Iceland response. While the surface and geochemical expressions of deglaciation enhanced melting are observed in Iceland, none of the basaltic flows in Yellowstone have been precisely dated to correspond to either recent deglaciation[32,35,36]. Moreover, even if deglacial basaltic flows are buried beneath newer material, modelled trace element depletions are within the range of existing observations, implying deglaciation signatures may not be resolvable in existing datasets (Fig. S14b). Thus, we infer that the processes governing melt migration

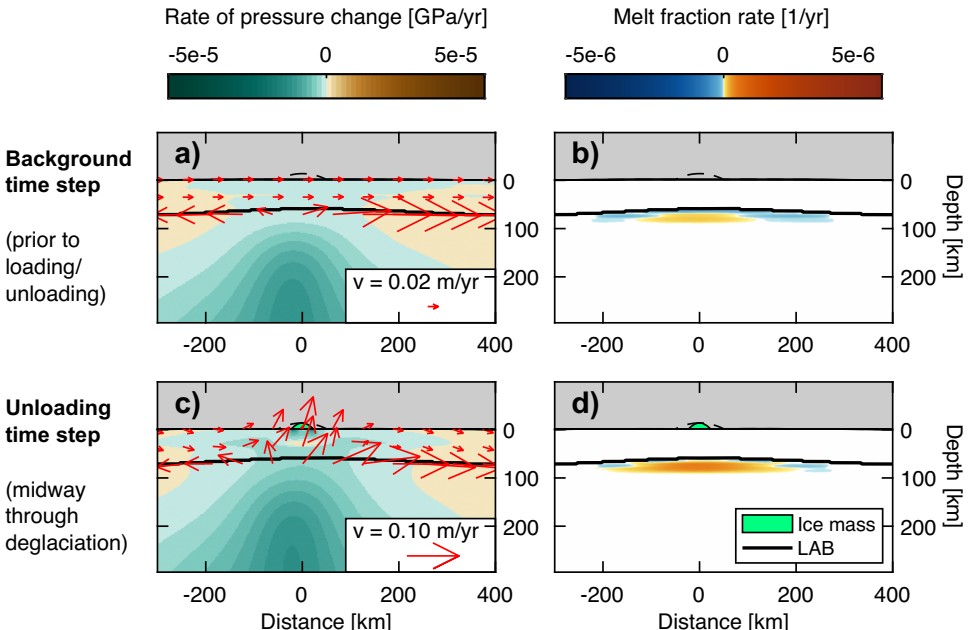

**Fig. 4 | Modeled melt production due to deglaciation of Yellowstone ice cap.** The ice cap is represented by the green parabola at a given time step and by the dashed black line at its maximum extent. Rates of pressure change are colored teal-brown (**a, c**) and rates of melt fraction change are colored blue-orange (**b, d**). The top row shows a model time step prior to any glacial loading/unloading, while the bottom row shows a time step 1000 years following deglaciation onset. Red arrows show mantle flow (at shallow depths only for ease of visualization), the thick black line is the lithosphere-asthenosphere boundary (LAB), at T = 1300 °C.

through the lithosphere and crust mitigate volcanic activity despite enhanced melting beneath Yellowstone. Understanding the transfer of the mantle melts to the surface is further complicated by the influence of unloading on the shallower magmatic system. Various studies have examined how magma chambers can be triggered by deglaciation[13,14]. Mantle melts may be pumped upwards as the continental lithosphere flexes during deglaciation[37]. We suspect that relative to Iceland, the thickness of the lithosphere beneath Yellowstone and the complexity of its magmatic system make it more difficult to efficiently transport mantle melts to the surface.

Even in the absence of anomalous eruption rates, large enhancements in mantle melting beneath Yellowstone can influence the crustal magmatic system. Bimodal basalt-rhyolite volcanism in Yellowstone may be explained by the co-existence of a rhyolitic upper crustal sill and a deeper basaltic reservoir[18]. The emplacement of mantle-derived melts into or near the upper crustal sill fuels rhyolitic eruptions, representing a source of heat and mass[38]. During the deglaciation we calculate an additional 360 km³ of mantle melt, which represents ~7% of the 5000 km³ of silicic melt estimated to be in the upper crustal sill today[18]. Similarly, the additional $2 \times 10^{21}$ J of heat we calculate could have been imparted to the sill during the deglaciation, sufficient to melt an additional 2900 km³ of near-solidus silicic melts, more than doubling the upper crustal sill volume. These upper-bound estimates illustrate that the emplacement of a large fraction of deglacial melts into or near the upper crustal sill may influence its dynamics or composition. Alternatively, the effect on the shallow magmatic system may be imperceptible, if for example the mantle melts travel slowly through the mantle and crust or are emplaced far from the sill.

The flux of $CO_2$ released to the surface by the crystallization of mantle melts at depth is less sensitive to upper crustal processes and may be the most consequential impact of deglaciation-enhanced melting beneath Yellowstone. Because we do not simulate melt transport, we resolve only the mass of excess $CO_2$ released, not the rate of degassing. However, we infer the release of an additional 18–79 Gt of $CO_2$ is likely not instantaneous (as might be implied by Fig. 3e), because $CO_2$ ascension will be slowed by magmatic and/or hydrothermal

processes. As a lower bound, if the additional $CO_2$ from Yellowstone is degassed over 20 kyr (implying melts travel through the lithosphere and lower crust at a rate of 2 m/yr), the enhanced flux would represent a ~0.3–1.5% increase in the global volcanic $CO_2$ flux[11]. In this scenario, the present-day Yellowstone flux may still be elevated by ~7–12 Mt/yr due to enhanced melting during the Pinedale deglaciation. Alternatively, if the enhanced $CO_2$ flux is degassed rapidly during a 2-kyr deglaciation, the enhanced flux would represent a ~3–15% increase in the global volcanic $CO_2$ flux[11] and could be accompanied by deglaciation-enhanced fluxes from other volcanoes, such as arcs[8,9]. The additional $CO_2$ from Yellowstone would increase the global deglacial $CO_2$ flux from active subaerial volcanoes since the Last Glacial Maximum[8] by 0.4–8%. For perspective, it has been proposed that the global deglacial $CO_2$ flux from arc volcanoes was responsible for the 40 ppm increase in atmospheric $CO_2$ between 13–7 ka (ref. 8). It is therefore possible that the enhanced release of magmatic $CO_2$ from Yellowstone also plays some role in this positive feedback between deglaciation and climate.

Another way in which deglaciation, climate warming, and volcanism may be linked is by the acceleration of ice flow due to volcanically enhanced geothermal heat fluxes (GHF). If heat associated with the emplacements of melt at depth was transported to the surface, it would be sufficient to melt 33% of the Yellowstone ice cap and 13% of the Iceland ice sheet. Large GHFs would maintain a thawed, water-saturated basal till and would soften the overlying ice, dynamically enhancing the mass loss of ice[39]. Yet in order to influence ice flow in Yellowstone, this additional heat must travel >10 km through the crust and reach the surface within the deglaciation interval (~1 kyr). The thermal conduction of heat from intruded basalts is negligible at ~kyr timescales[40]. Instead, advective heat transfer would require mass fluxes of magmatic and hydrothermal fluids of >10 m/yr to affect ice dynamics during the deglaciation interval. We note that the timing of deglaciation in both Iceland Yellowstone corresponds to climate warming (e.g., refs. 15,41), implying glacier retreat is driven primarily (and necessarily initiated) by external climatic factors. In numerical simulations of the retreat of the Iceland ice sheet, enhancing the GHF by 50% (relative to present-day values) minimally influences ice flow[41].

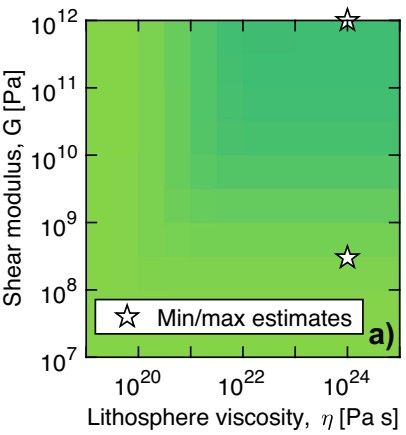
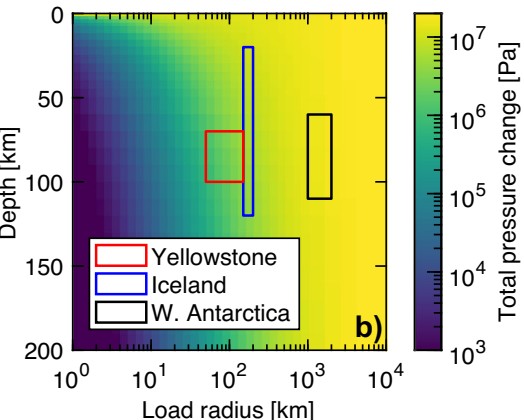

**Fig. 5 | Influence of rheologic and loading parameters on change in pressure at depth.** (yellow-green colors) after one deglaciation, in a viscoelastic half-space. A pressure change of $10^6$ Pa in partially molten mantle scales to 3 km³/km² of additional melt, under a melt productivity (dF/dP) of 0.15/GPa and 20-km melt column. Pressure changes as a function (**a**) shear modulus (G) and overlying viscosity, at a constant depth of 70 km and load radius of 50 km. The influence of rheology is small, and is bounded in the limit of rigid/soft lithosphere (white stars correspond to the parameter analysis in Section S3.4). **b** For constant rheological parameters (G = 10 GPa, η = $10^{24}$ Pa s), the load radius controls the magnitude of pressure changes as a function of depth (beneath the load center). Rectangles represent different settings (red for Yellowstone, blue for Iceland, black for West Antarctica). Note that pressure changes are directly proportional to load height, here kept fixed at 1.25 km.

Yet given the colocation of paleo ice streams and geothermal features in Iceland[42], the effect of a larger (as estimated here) and more localized GHF enhancement remains to be explored. Beneath Yellowstone, rising melts may induce a response in the hydrothermal system by imparting heat[43] or $CO_2$ (ref. [44]). In fact, larger hydrothermal explosion craters are observed during the last glaciation[45], although hydrothermal explosions also occurred throughout the Holocene, unrelated to deglaciation[46,47]. The reactivation of faults due to deglaciation[48] conceivably also influences hydrothermal fluid flow.

### Implications for West Antarctica and Greenland

Placing our findings in a broader context, we suggest magmatically-active continental systems may experience enhanced mantle melting in response to deglaciation. To quantify this feedback, we examine the influence of lithosphere parameters on the total change in pressure at depth, following a deglaciation (Fig. 5). We use the semi-analytical, radially-symmetric solution of JM96 for a viscoelastic half-space and compare this scaling with our numerical results (Section S3.4). Under a given loading history, the influence of rheological parameters (shear modulus, mantle viscosity) on enhanced melting is limited, changing the initial model result by 0–13% (Fig. S11a). By contrast, the load radius exerts an important control on the depth to which changes in pressure are felt (Fig. 5b). The smaller size of the Yellowstone ice cap (red box, Fig. 5b) results in an order-of-magnitude reduction in deglacial melt volume, relative to the Iceland case (blue box). Further, varying the volume of the melt region, the thickness of the ice load, and the dependence of the solidus on pressure scales the deglacial melt volumes proportionately (not shown in Fig. 5b). Under large load radii (>1000 km) – i.e., the retreat of large continental ice sheets – pressure changes barely decay at mantle melting depths (black box, Fig. 5b).

Moreover, deglaciation may enhance transient melting anomalies that would not be otherwise productive, supporting the idea that, if present, remnant melts beneath Greenland may be influenced by deglaciation[4]. The transient melting anomaly model (Fig. S8) implies deglaciation can enhance melting in the upper mantle over a range of geodynamic conditions, in settings characterized by a partially molten mantle.

In particular, West Antarctica is volcanically active[49] and characterized by relatively thin (60–110 km) lithosphere[50]. Other tectonic similarities between the West Antarctic Rift System (WARS) and Yellowstone include the possible existence of a mantle plume[51] and extensional lithospheric stresses. During some interglacials, paleo proxies suggest the collapse of the West Antarctica Ice Sheet (WAIS) (ref. [52]), and models predict the loss of millions of km³ of ice over short (-kyr) timescales[53]. The horizontal extent of the WAIS also implies deglacial unloading will generate larger rates of decompression at asthenospheric depths compared to our calculations for Yellowstone (Fig. 5b). Finally, while the total flux of $CO_2$ from West Antarctic volcanism is unconstrained, other continental rift systems are important $CO_2$ emitters[11] and in some locations the Antarctic mantle is rich in $CO_2$ (ref. [54]). Thus, melt production rates and associated $CO_2$ fluxes released into the atmosphere may be greatly enhanced under WAIS collapse and could drive a positive feedback with climate warming. As modern elevated GHF already influence ice flow[5–7], deglacially enhanced melting may further impart heat to the base of the WAIS and accelerate its collapse. Understanding the magnitude of deglacially enhanced melting beneath West Antarctica will require further model development, and has implications for global carbon budgets, climate, and the evolution of the WAIS over millennial time scales.

## Methods

We examine deglaciation-enhanced mantle melting beneath Iceland and Yellowstone using the mantle convection code ASPECT[21,22]. The models are sufficiently idealized to facilitate comparison between both settings, yet capture key geodynamic differences and match various observations. We estimate $CO_2$ and heat fluxes to understand the surface impact.

### Model initialization

The mantle is assumed to behave as a Newtonian visco-elasto-plastic material with a temperature-dependent viscosity. Viscosities are calculated for dry dislocation creep[55], assuming an activation volume of $20 \times 10^{-6}$ m³/mol, an activation energy of 500 kJ/mol, and a pre-factor $1.1 \times 10^5$ MPa⁻²·⁵ s⁻¹. Viscosities are converted to a Newtonian form, using an effective deviatoric stress of 1.2 MPa for Iceland, and 0.7 MPa for Yellowstone. These parameters yield asthenospheric viscosities of $0.5–1.0 \times 10^{19}$ Pa s in the absence of a plume thermal anomaly. Elasticity is characterized by a shear modulus of $10^{10}$ Pa. A Mohr-Coulomb failure law allows rapid deformation at the Iceland ridge axis during spin-up, otherwise plasticity is not activated.

Mantle potential temperatures of 1300 °C for Iceland and 1320 °C for Yellowstone are assumed in the absence of a plume. Plumes are

initiated with a thermal Gaussian anomaly at 600 km depth, centered at x = 0 km (Fig. 1). Plume excess temperature and radius at 600-km depths are 175 °C and 100 km for Iceland and 80 °C and 70 km for Yellowstone, respectively, in accordance with previous work benchmarked against geophysical observations[25,56–58]. The plume underneath Iceland is centered beneath a symmetrical ridge axis, while the Yellowstone plume impinges upon a moving plate, such that the model is asymmetrical. During model spin-up, the top boundary condition is driven by plate motions (10 mm/yr for Iceland, 20 mm/yr for Yellowstone). The remaining boundaries are open, with the exception of the free-slip symmetry condition at the Iceland ridge axis. The models are run until the thermal structure and flow field stabilize (10–30 Myr).

### Modeling the response to glacial unloading

The models are unloaded by decreasing the ice sheet radius at a constant rate over a prescribed deglaciation interval (1000 years for Iceland, 2000 years for Yellowstone), simulating the retreat of the ice margin. The flow through the open boundaries is then fixed to the steady-state value, and the top boundary becomes a free surface that deforms in response to applied pressures. After the glacial load is applied, the model is again allowed to stabilize to rule out the influence of the glaciation. The Iceland ice sheet is simulated as a parabola 180 km in radius and 2 km high (as in JM96). While JM96 kept the load radius constant and horizontally thinned the ice sheet thickness, we assume the load retreats laterally (vertically) from the margins. We compare the horizontally thinned load from JM96 (constant radius, decreasing thickness), the vertically retreating load (decreasing radius, constant maximum thickness) shown in Fig. 2, and a horizontally and vertically retreating smaller load (following refs. 41,59). In vertically retreating simulations, the melt production rate increases through time as the zone of maximum decompression migrates towards the ridge axis where the load is centered (Figure S4). For the Yellowstone ice cap, we use an elliptic paraboloid with a short axis of 50 km (trending E-W, parallel to plate motion) and a height of 1.25 km. We run a 2-D Cartesian model along the short axis, and a 3-D model with a long axis of 150 km (see Section S3.5). This yields a volume of 15,000 km³ (ref. 15) and an average height of ~600 m (Pierce[60] estimates 700 m from inferred basal shear stresses). Assuming the margins of the ice cap retreat at a constant rate, 3-D effects may imply the 2-D results overestimate melt fluxes by ~10% (Fig. S12). Unloading the ice cap horizontally instead of vertically does not influence melt production rates (Fig. S9a). The dimensions of the Yellowstone ice cap correspond to the most recent and well-constrained Pinedale deglaciation (15–14 ka), we assume the more relevant penultimate Bull Lake glaciation (~150 ka) retreated similarly. Lengthening the duration of the deglaciation reduces the melt production rate; however, the total volume of melt produced over the entire deglaciation is unchanged and depends solely on the volume of ice lost (Fig. S9b).

The rate of melt fraction change depends on the material derivative of the pressure field, which includes both instantaneous (elastic) changes in pressure and isostatic rebound. We also include the dependence of the melt fraction rate on the temperature field due to the effects of latent heat (as in ref. 61). We use a dry peridotite solidus[62], implying our models underestimate melt volumes under hydrated mantle conditions. Melt fractions (Fig. 1c, d) and their dependence on pressure/temperature remain relatively constant through time as the deglaciation time scales are short. To obtain melt production rates, we spatially integrate positive rates of melt fraction change (excluding regions of freezing).

### Iceland model

The Iceland melt production rate is presented in 2-D. To estimate 3–D volumetric melt production rates and CO₂ fluxes, we integrate along a 100-km-long portion of the ridge axis (half the plume head width over which the temperature structure stays relatively constant[25]). We

caution that this scaling is approximate and is only useful as a first-order comparison against the Yellowstone models. Our 2-D slice through the plume center represents the maximum melt production, such that the total 3-D rate would average with less productive regions away from the plume center. Comparisons to the JM96 benchmark are presented in 2-D. The total background melt production rate over the entire model domain of 0.23 km³/yr is equivalent to a steady-state crustal thickness of 128 km for a 20 mm/yr full-spreading rate and mantle and crustal densities of 3000 and 2700 kg/m³, respectively. This is higher than the observed crustal thickness of 20–40 km (refs. 63,64). As melts generated far from the ridge axis likely refreeze at the base of the lithosphere[23,24,65], we assume only melts generated within 30 km of the axis contribute to the production of crust (see Section S2.3). This pooling width yields 39 km of crust, consistent with seismic observations of the Iceland crust[63,64]. We ran the same model without the plume and obtain a crustal thickness of 7 km, typical of slow-spreading mid-ocean ridges[66].

### Yellowstone model

The 3-D melt production rate for Yellowstone is calculated by radial integration of melt fraction rates. We benchmark this calculation using a full 3-D model (Section S3.5, Fig. S12). The maximum melt fraction beneath Yellowstone is 3.5% and the mantle potential temperature (including the excess plume temperature) is 1400 °C, consistent with geophysical and geochemical constraints[31,67]. The absence of melts at depths >90 km in our model is attributable to the use of a dry solidus. The melts must leave the asthenosphere rapidly to avoid refreezing in the outer wings of the melt region, which reaches depths of 60 km at the shallowest point (Fig. 4b; blue colors). The background melt production rate of 0.022 km³/yr is comparable to the estimated emplacement rate of basalts into the crust (0.005–0.025 km³/yr) based on uplift rates and thermal arguments[20,32]. The simulation in which the plume tail is removed has a smaller background melt production rate of 0.006 km³/yr, due to the absence of uplift from the lower mantle. Following alternative estimates derived from chloride[68] and CO₂ flux[69] considerations, we also vary the mantle temperature (including the excess plume temperature) to 1420 °C and 1440 °C and obtain melt production rates of 0.056 and 0.15 km³/yr, respectively (Fig. S6). The simulation with the 1420 °C mantle temperature produces an extra 615 km³ of melt (representing a 6-fold enhancement) and 10–48 Gt of CO₂. The simulation with the 1440 °C mantle temperature produces an extra 962 km³ of melt (representing a 4-fold enhancement) and 3–24 Gt of CO₂. Under the most productive conditions, the extra CO₂ released is smaller as we must assume lower source mantle CO₂ concentrations to match modern CO₂ fluxes (Fig. S6b).

### Calculating concentrations of CO₂, trace elements, and heat flux

For both Iceland and Yellowstone, we calculate trace element concentrations using a non-modal retained batch melting formulation[26], assuming partition coefficients for peridotite melting[70] and depleted mid-ocean ridge basalt (MORB) mantle source concentrations[71]. In the limit that all melt is retained, the element concentrations approximate that of a batch melt; conversely if no melt is retained the solution approximates fractional melting. We use a retained melt fraction of 1 wt.% as it best fits available data (Fig. S15). Further, this value is within the range of porosities (0.2–1.95 wt.%) that explain the reduction in seismic velocity in Yellowstone's asthenosphere (ref. 31), and is within the range (0.1–1 wt.%) consistent with geochemical signatures in MORBs[72,73]. The element concentrations in the pooled melts are weighted by the melt production function (as in refs. 1,59), and vary during unloading. During the deglaciation of Iceland, trace element concentrations provide evidence that mantle melting was enhanced, as incompatible light rare Earth elements (LREE) become more diluted under greater melting rates[1,2,27]. We compare the percent change in the LREE compositions between the unloading period and a background

time step (Fig. S3). While our method and partition coefficients differ from those used by JM96, we obtain similar changes before and after unloading in our benchmark case with otherwise identical assumptions and parameters (-15% change for lanthanum).

Using the same approach and assuming $CO_2$ partitions into the melt similarly to barium[74], we estimate the flux of $CO_2$ segregated from the mantle by melts. If the melts are emplaced at depth, greater lithostatic pressures imply increased solubility of $CO_2$ in the melt. For Iceland, we assume the melts generated near the ridge axis (within the 30-km "pooling width") are erupted or emplaced at shallow depths such that the $CO_2$ is perfectly outgassed to the surface. By contrast, we assume that melts generated off-axis are emplaced locally at the base of the lithosphere (defined as the 1300 °C isotherm), such that some of the $CO_2$ remains dissolved in the melts, depending on the lithostatic pressure (following ref. 75). We explore the influence of pooling width on crustal production and $CO_2$ flux in Section S2.3. Melt inclusion compositions[76] indicate the bulk concentration of $CO_2$ in the Icelandic mantle is a mix of a deep mantle component containing -1350 ppm $CO_2$ and a depleted mantle component containing -120 ppm $CO_2$ (ref. 77). To simulate different mixtures of these components, we show results for mantle source concentrations of 150 and 600 ppm $CO_2$ in Fig. 4d. For Yellowstone, we assume the melts crystallize at 14 km, the base of the upper crustal sill[18]. This implies 0.25 wt.% $CO_2$ is retained in carbonate form[75], such that 80% of the $CO_2$ segregated from the mantle is released to the surface. We compare the flux of $CO_2$ exsolved to the surface with published estimates of $CO_2$ released into the atmosphere by magmatic activity (Fig. 3d, e). While we explore different mantle source $CO_2$ concentrations, we do not model the effect of these different concentrations on the degree of melting. Omission of low-degree carbonate melting does not affect melt volumes substantially, but could cause underestimates in $CO_2$ fluxes.

We use the melt production rates to estimate geothermal heat fluxes. We assume the basaltic mantle melts have a density of 2800 kg/$m^3$, specific heat of 1500 J/kg/K, and latent heat of 400 kJ/kg (ref. 78). From our numerical model, we obtain the difference in temperature between the depths of melt generation and emplacement. For Iceland, we consider the emplacement of melts at a depth of 10 km (ref. 30) and assume the melts are 300 °C warmer than the surrounding crust. The temperature difference is estimated by comparing modelled temperatures at the locus of greatest melt fractions against temperatures at the depth of emplacement (corrected for adiabatic cooling). For Yellowstone, we consider the emplacement of all the melts near the base of the upper crustal sill (-14 km), and assume that the melts are 1000 °C warmer than the surrounding crust. We estimate the specific and latent heat released as melts cool and crystallize. We scale these heats by the emplacement rate, to estimate the total heat imparted by the melts at the depth of emplacement. We also assume silicic melts have a latent heat of 300 kJ/kg and density 2300 kg/$m^3$ (ref. 78), and ice has a latent heat of 334 kJ/kg and density 900 kg/$m^3$.

## Data availability
No datasets were generated during this study.

## Code availability
The code which generates our results is available at <https://github.com/fionaclerc/deglaciation_melting>, and at the following repository: 10/5281/zenodo.10529699 (ref. 79).

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

## Acknowledgements
We thank Greg Hirth, Véronique LeRoux, Brad Hager, Shaul Hurwitz, and Lauren Harrison for their helpful comments. We thank John Naliboff, Juliane Dannberg, Garrett Ito, Daniel Douglas, and Maaike Weerdesteijn for help with the benchmarking of the numerical model. Computations were performed on WHOI's Poseidon HPC, sponsored by Dan Lizarralde. We acknowledge funding from NSF grants OCE-14–58201 (F.C.), OPP-18-38410 (M.B.), and an NSF GRFP (F.C.).

## Author contributions
F.C. led the study, performed the computations, and drafted the manuscript. M.B and B.M. advised on study design and revised the manuscript.

## Competing interests
The authors declare no competing interests.

## Additional information
**Supplementary information** The online version contains Supplementary Material available at https://doi.org/10.1038/s41467-024-45890-z.

