## [Peer Review File · Nature Communications]

Deglaciation-enhanced mantle CO₂ fluxes at Yellowstone imply positive climate feedbackReviewer #1 (Remarks to the Author):

This is a fascinating paper that makes an important and novel exploration of the effect of glacial unloading on melt generation in continental intra-plate magmatic settings. The scientific target is important and this manuscript is a welcome first attempt to model such settings. The authors bench-mark their method against the well characterised and previously modelled case of Iceland. They then use this method to explore melt generation associated with the Yellowstone area. Notably, their models predict a significant enhancement in the rates of mantle melting as a result of glacial unloading in this area. Unfortunately, there is relatively little data available to test these models at present.

There are some important improvements that could be made to this manuscript. There is a great deal of model development here, and I found myself having to switch between the main text, methods text, supplementary text and figures at almost every step. I wonder if the manuscript would benefit from being in a longer format? The methods section would benefit from being split into clear sections.

I also found some of the model description and results a bit puzzling and hope that the authors can clarify certain portions of the text:

1. Overall melt generation at Iceland (L97 in main text, L339-348 in methods section).

I found the arguments here confusing. First, it would be good to compare the melt generation from the Iceland model with the relatively well-constrained steady-state melt production for the island (full spreading rate about 20 mm yr⁻¹, about 300 km of rift zone, and average crustal thickness of about 30 km in the rift zones – See Jenkins et al, JGR, 2018, for a recent look at crustal thicknesses). This gives a little under 0.2 km³ per year from the subaerial portions of the spreading ridge near Iceland. This is not too dissimilar from their calculated background melt production rate of 0.115 km³/yr – but it would have been useful to make the direct comparison. I was therefore mystified to see the calculations of a mean steady-state crustal thickness of 128 km! – why would a 10 mm yr⁻¹ spreading rate and a 100 km along-axis width be appropriate here? Surely the relevant lengthscale is the portion of the ridge that is exposed above sea-level and was glaciated – which is closer to 300 km, and the relevant spreading rate is the full spreading rate (not the half-spreading rate which would make sense in the model calculation. These calculations were confusing for me.

2. Trace element calculations (L125 main text, L367-380 in methods).

It would be very helpful to have some equations to clarify exactly what was done here, and to contrast this approach with the calculations in JM96 and Eksinhol. On L370 they say that the “pooled melts” are weighted by the “melt production function” but I worried whether this was equivalent to the weighting of an instantaneous fractional melt composition by the melting rate at the point of melt production (or escape from the residue in their case). Based on the description provided in the manuscript, I felt that I would have a hard time building a model to replicate their findings.

3. Sensitivity to mechanical model parameters.

One important model result is that enhanced mantle melt production can occur in locations with relatively thick lithosphere (~100 km). The manuscript could be improved by further testing of that finding. In L75 we are told that ASPECT was used and then in the methods on L297-302 we are provided with information about the mechanical set-up. How robust are the findings to the particular details of the code that was used and the parameter values that the authors chose here? Can some simple approximations be used to demonstrate that important decompression can be transmitted through thick

lithosphere to the underlying melting region? This is a crucial point and requires attention.

4. Excess heat production (L135, L193, L247 main text)

My feeling is that the authors spend too much time speculating on this problem – I think that they can calculate the excess heat associated with the burst in melt production and state that. They recognise that for this heat to influence the ice-sheet it is necessary for it to be supplied rapidly to shallow levels, either directly by subglacial eruption or by hydrothermal convection. Given the huge uncertainties here, I would suggest that the amount of text dedicated to this aspect of the model results is reduced.

5. Calculation of enhanced melting rates.

In Figure 4 it is clear that the steady-state models involve a prediction of zones where compression is occurring a freezing is predicted. I wondered if these freezing regions (melt fraction rate <0) were included in the overall steady-state melt production calculation and the comparison or steady-state and unloading melt production rates. I would have thought that only points with melt fraction rate > 0 should be included, but was not clear on that point. This could make a difference to the reported enhancement in melt rates.

Reviewer #2 (Remarks to the Author):

The manuscript under review presents results of computational modelling of melting beneath Iceland and Yellowstone during the last deglacial period. Melting was enhanced during this period, which is captured by the models presented. The noteworthy result is a prediction of an extra $\sim 200\text{Gt}$ of CO_2 from the mantle delivered to the climate system during deglaciation. This prediction is of broad significance because of its implications for climate. In particular, it would represent a positive feedback on climate warming at the end of the last ice age.

Previous workers have addressed the effect of deglaciation on melt production, and also on CO_2 release from Iceland during deglaciation. This work is novel in applying the same theory to Yellowstone, a major mantle hotspot.

The authors have done a nice job of modelling the mantle dynamics of Iceland and Yellowstone. They use an appropriate rheology, boundary conditions, and deglacial forcing. Their calculations of melt production in Iceland compare well with widely accepted results from previous work that agree qualitatively with observations. The arguments regarding melt and CO_2 are plausible, but I find the models less convincing. This may be due to a somewhat limited explanation: the authors rely heavily on the output of the numerical models to make their point and don't include any simple scaling analysis to support the code output. Lacking a simple and transparent means to support their results, I have a few concerns that I think should be addressed.

A quantitative accounting for CO_2 should be based on equations for conservation of mass. This might take the form of a single equation, if the CO_2 concentration is assumed to be in equilibrium, or two equations for the solid and liquid phases if not. The Armitage study cited by the authors is an example, but a more rigorous combination of melt segregation and mass conservation of volatiles was presented by Cerpa et al EPSL 2019. The reason that this is important, especially in the case of Yellowstone, is that the mantle moves very slowly -- it doesn't replenish the melting region with solid material over the 100ka glacial cycle (indeed at 0.1 m/yr peak upwelling, the mantle moves upward only 10km or less over this cycle, but this rate is achieved only during deglaciation, which is a small fraction of the total period). Melting removes effectively

all of the CO₂ from the solid phase ($D \sim 1e-3$ or $1e-4$) and so mantle that was previously melting should have lost almost all of its CO₂. I think the authors must give a clear explanation of how they conserve mass of carbon in this two-phase problem.

This is where my second concern comes in. The authors don't really use a melt-transport model, but rather a "retained batch melting formulation" with a retained melt fraction of 1%. It is unclear to me whether such a model can be applied to a non-steady problem (this problem is non-steady because of the deglacial forcing). Furthermore, it is unclear what the assumption of a 1% retained melt fraction has. This number seems somewhat arbitrarily chosen and I suspect it has a very big effect on the rate of CO₂ release. A smaller retained fraction might lead to a sharper peak in CO₂ transport. I think the authors must explain, justify, and explore their use of the retained batch melting formulation, and provide some sensitivity analysis to the chosen 1% factor. For reproducibility, I think a more detailed explanation of the melt-transport scheme is also required.

The combination of the above two concerns leads to my third concern, which is that the present deglaciation is affecting mostly the same mantle rock that was beneath Yellowstone for the previous deglaciation, because that mantle is upwelling rather slowly compared to Iceland. So where is the excess CO₂ sourced from? The bottom of the deepening melting regime, as hypothesised by Burley & Katz? Or simply by accelerating the segregation of the melt that was already in the melting region? Cerpa et al concluded that for MORs, the latter is much more important than the former. But this depends on the model for melt segregation, and also on the melting history of the solid mantle beneath Iceland. The authors seem to assume in their calculations that the mantle had previously undergone only steady upwelling, whereas they discuss the penultimate deglaciation being relevant for Yellowstone. I think a model of oscillatory glacial load would be a very nice addition, though this might be too much to ask. In any case, the authors should give some discussion and scaling analysis of the source of the excess CO₂ and the sensitivity of this to their assumptions. A corollary concern is that the mantle, having been depleted by the previous melting event during the previous deglacial, is now more refractory than assumed.

The authors do provide comparison between the excess CO₂ delivered in their models with previous estimates. The rough agreement is taken as an implied validation of the models, but this may be misleading. Estimates of CO₂ degassed during the last deglacial may be dependent on the same assumptions made here.

Other minor comments:

- Comparison of figures 2 and 4. Why are there no velocity vectors in the asthenosphere of figure 4? Why is the decompression colour scale the same as in figure 2? Why are the red arrows in the legend of panels (a) so small? Does this represent the scale of the flow? If so, it makes the figure more difficult to understand.

- Figure 3. Why is the total CO₂ rate perturbation larger for Yellowstone than for Iceland, and also greater over a longer period of time (2ka versus 1ka) and yet the estimates for the total CO₂ excess is 135--230 Gt in both cases (in the text)? These settings are very different, so shouldn't we expect rather different outcomes of deglaciation? If not, this would seem to call for a careful explanation, which seems to be missing.

- Lines 152-158 need references.

- I found the discussion of dimensionality of models and results confusing. The models seem to be 2D Cartesian, which is a bit awkward for a plume model. But the authors seem to make some sort of 3D correction. I found all of this unclear.

- What is the definition of the "melt fraction rate"? I could understand the degree-of-

melting rate, because this is a property of the solid phase. Melt fraction is, in my mind, equal to porosity, which is a property of the two-phase system and therefore affected by both melting and segregation. So I don't see how it can be used as a proxy for melting alone.

- I liked the energetic comparisons that lead to the idea that magmatic heat transfer could contribute to ice melting and acceleration, but if this relies on excess melting due to deglaciation then the argument might be a bit circular...

We thank the reviewers for their comments, which have improved the quality of the manuscript. We have written our responses below.

We also note that in the time between initial submission of the manuscript and revisions, Rahilly & Fischer (2021) published updated estimates of modern Yellowstone CO₂ fluxes, which are half of the original estimates from Werner & Brantley (2003). We have updated our model to reflect this.

Line numbers refer to the “tracked changes” version of the manuscript. We note we have edited the abstract to fit the word limit.

Reviewer #1 (Remarks to the Author):

This is a fascinating paper that makes an important and novel exploration of the effect of glacial unloading on melt generation in continental intra-plate magmatic settings. The scientific target is important and this manuscript is a welcome first attempt to model such settings. The authors bench-mark their method against the well characterised and previously modelled case of Iceland. They then use this method to explore melt generation associated with the Yellowstone area. Notably, their models predict a significant enhancement in the rates of mantle melting as a result of glacial unloading in this area. Unfortunately, there is relatively little data available to test these models at present.

There are some important improvements that could be made to this manuscript. There is a great deal of model development here, and I found myself having to switch between the main text, methods text, supplementary text and figures at almost every step. I wonder if the manuscript would benefit from being in a longer format? The methods section would benefit from being split into clear sections.

As suggested, we have split the methods section into titled sections. We have also expanded the supplementary material (adding sections S3.4, S3.5, S5), with the intention of clarifying details of model development and providing insight into model results and their sensitivity to model parameters.

I also found some of the model description and results a bit puzzling and hope that the authors can clarify certain portions of the text:

1. Overall melt generation at Iceland (L97 in main text, L339-348 in methods section).

I found the arguments here confusing. First, it would be good to compare the melt generation from the Iceland model with the relatively well-constrained steady-state melt production for the island (full spreading rate about 20 mm yr⁻¹, about 300 km of rift zone, and average crustal thickness of about 30 km in the rift zones – See Jenkins et al, JGR, 2018, for a recent look at crustal thicknesses). This gives a little under 0.2 km³ per year from the subaerial portions of the spreading ridge near Iceland. This is not too dissimilar from their calculated background melt production rate of 0.115 km³/yr – but it would have been useful to make the direct comparison. I was therefore mystified to see the calculations of a mean steady-state crustal thickness of 128 km! – why would a 10 mm yr⁻¹ spreading rate and a 100 km along-axis width be appropriate here? Surely the relevant lengthscale is the portion of the ridge that is exposed above sea-level

and was glaciated – which is closer to 300 km, and the relevant spreading rate is the full spreading rate (not the half-spreading rate which would make sense in the model calculation. These calculations were confusing for me.

We recognize that reporting the 3-D volumetric melt production rates is confusing given that our models for Iceland are in 2-D, and have made the conversion more explicit by reporting both the volumetric flux (assuming a length-scale) and length-normalized flux in the text (e.g., lines 106–07, 116, etc.).

Regarding the use of a half-spreading rate, we do this as we only model half of the ridge and assume a symmetry condition at the left boundary (multiplying our melt production rate by 2 and then dividing by the full spreading rate is the same as simply dividing by the half-spreading rate). Thus our crustal thickness value is accurate, but in reporting the full 3-D melt production rate (and additional deglacial melt volumes/CO₂ masses) we forgot to multiply by 2, to account for the other side of the ridge. This is corrected on line 398. It also implies that the excess melt volume, mass of CO₂, and heat fluxes must be doubled (updated on lines 116, 141, 149, 153, etc).

We chose to multiply by a 100-km along axis length-scale for Iceland, as this is the plume diameter over which the temperature structure stays relatively constant in 3-D models (e.g., Ito et al., 1999). We do not consider contributions further away from the plume head. We do agree that the melt produced in our model cannot be emplaced within this 100-km zone nearest to the plume head, as this would imply crustal thicknesses that are too large. We expect this extra material is emplaced laterally along the rift zone (see lines 400–01). If most of the material is produced within 100-km of the plume head and emplaced within 300 km, the 128-km crustal thickness would be ~40 km, consistent with seismic observations (e.g., Jenkins et al., 2018). We clarify this on line 403–405.

2. Trace element calculations (L125 main text, L367-380 in methods).

It would be very helpful to have some equations to clarify exactly what was done here, and to contrast this approach with the calculations in JM96 and Eksinhol. On L370 they say that the “pooled melts” are weighted by the “melt production function” but I worried whether this was equivalent to the weighting of an instantaneous fractional melt composition by the melting rate at the point of melt production (or escape from the residue in their case). Based on the description provided in the manuscript, I felt that I would have a hard time building a model to replicate their findings.

We use a similar approach as JM96 and Eksinhol et al. (2019) in which the instantaneous melt compositions are weighted by cumulative melt production functions and summed over the melt path. The difference in our approach is in the partitioning of the trace elements into the melt. We clarify these points and describe these methods in greater detail on lines 431–37. We have uploaded our codes to Github (https://github.com/fionaclerc/deglaciation_melting) to aid reproducibility (stated lines 478–80).

3. Sensitivity to mechanical model parameters.

One important model result is that enhanced mantle melt production can occur in locations with relatively thick lithosphere (~100 km). The manuscript could be improved by further testing of that finding. In L75 we are told that ASPECT was used and then in the methods on L297-302 we

are provided with information about the mechanical set-up. How robust are the findings to the particular details of the code that was used and the parameter values that the authors chose here? Can some simple approximations be used to demonstrate that important decompression can be transmitted through thick lithosphere to the underlying melting region? This is a crucial point and requires attention.

We agree that this is a crucial point and have expanded our discussion (lines 289–302; Figure 5) as well as the supplementary information (Section S3.4; Figures S9 & S10) to address it. In the revised text we use the semi-analytical solution (JM96) to examine the conditions necessary to transmit pressure changes through thicker lithosphere (Figure 5). The length scale of pressure decay (with depth) depends on the width of the glacial load (e.g., Haskell, 1935), with lesser sensitivity to the rheological parameters.

In addition, we benchmark our results against the semi-analytical solution of JM96 (Figure S1) – and thus expect our results are robust regardless of the numerical method. We also show the influence of different assumptions and additional (physically-based) complexities in our models, in Figure S2.

4. Excess heat production (L135, L193, L247 main text)

My feeling is that the authors spend too much time speculating on this problem – I think that they can calculate the excess heat associated with the burst in melt production and state that. They recognise that for this heat to influence the ice-sheet it is necessary for it to be supplied rapidly to shallow levels, either directly by subglacial eruption or by hydrothermal convection. Given the huge uncertainties here, I would suggest that the amount of text dedicated to this aspect of the model results is reduced.

This is a good suggestion. We have softened the language/reduced text on heat production on lines 275–77, 2880, 282–84.

5. Calculation of enhanced melting rates.

In Figure 4 it is clear that the steady-state models involve a prediction of zones where compression is occurring a freezing is predicted. I wondered if these freezing regions (melt fraction rate < 0) were included in the overall steady-state melt production calculation and the comparison or steady-state and unloading melt production rates. I would have thought that only points with melt fraction rate > 0 should be included, but was not clear on that point. This could make a difference to the reported enhancement in melt rates.

Yes, in our summation, we only include areas with a positive melt fraction rate. This is now clarified on lines 391–92.

Reviewer #2 (Remarks to the Author):

The manuscript under review presents results of computational modelling of melting beneath Iceland and Yellowstone during the last deglacial period. Melting was enhanced during this

period, which is captured by the models presented. The noteworthy result is a prediction of an extra ~200Gt of CO₂ from the mantle delivered to the climate system during deglaciation. This prediction is of broad significance because of its implications for climate. In particular, it would represent a positive feedback on climate warming at the end of the last ice age.

Previous workers have addressed the effect of deglaciation on melt production, and also on CO₂ release from Iceland during deglaciation. This work is novel in applying the same theory to Yellowstone, a major mantle hotspot.

The authors have done a nice job of modelling the mantle dynamics of Iceland and Yellowstone. They use an appropriate rheology, boundary conditions, and deglacial forcing. Their calculations of melt production in Iceland compare well with widely accepted results from previous work that agree qualitatively with observations. The arguments regarding melt and CO₂ are plausible, but I find the models less convincing. This may be due to a somewhat limited explanation: the authors rely heavily on the output of the numerical models to make their point and don't include any simple scaling analysis to support the code output. Lacking a simple and transparent means to support their results, I have a few concerns that I think should be addressed.

In response to this comment and a similar one from Reviewer #1 (concerning mechanical properties of the lithosphere), we have expanded our discussion (lines 289–302), added Figure 5, and expanded the supplement (Section S3.4; Figures S9 & S10). One simple scaling argument that we now highlight is that the load radii (50–150 km; see discussion on 2–D vs 3–D in Section S3.5) is similar to the depth of melting of ~70 km – implying pressure changes penetrate to these depths (Figure 5b; Figure S9). To illustrate this point, we input the parameters for Yellowstone into the semi-analytical solution from JM96 (Figure 5), accompanied by further sensitivity analyses in the numerical models (Figure S10).

A quantitative accounting for CO₂ should be based on equations for conservation of mass. This might take the form of a single equation, if the CO₂ concentration is assumed to be in equilibrium, or two equations for the solid and liquid phases if not. The Armitage study cited by the authors is an example, but a more rigorous combination of melt segregation and mass conservation of volatiles was presented by Cerpa et al EPSL 2019. The reason that this is important, especially in the case of Yellowstone, is that the mantle moves very slowly -- it doesn't replenish the melting region with solid material over the 100ka glacial cycle (indeed at 0.1 m/yr peak upwelling, the mantle moves upward only 10km or less over this cycle, but this rate is achieved only during deglaciation, which is a small fraction of the total period). Melting removes effectively all of the CO₂ from the solid phase ($D \sim 1e-3$ or $1e-4$) and so mantle that was previously melting should have lost almost all of its CO₂. I think the authors must give a clear explanation of how they conserve mass of carbon in this two-phase problem.

Our model assumes that the melt fractions (degree-of-melting) and CO₂ content of the mantle does not deviate strongly from the steady-state values. We agree that accounting for the transient depletion of the mantle due to deglacial melting (and coupling with the evolution of the melt system) would be more rigorous, but we estimate that this effect is small in comparison with uncertainties surrounding the background CO₂ flux at Yellowstone. We make this point with the new Section S5, which includes conservation of mass of CO₂, as suggested.

This is where my second concern comes in. The authors don't really use a melt-transport model, but rather a "retained batch melting formulation" with a retained melt fraction of 1%. It is unclear to me whether such a model can be applied to a non-steady problem (this problem is non-steady because of the deglacial forcing). Furthermore, it is unclear what the assumption of a 1% retained melt fraction has. This number seems somewhat arbitrarily chosen and I suspect it has a very big effect on the rate of CO₂ release. A smaller retained fraction might lead to a sharper peak in CO₂ transport. I think the authors must explain, justify, and explore their use of the retained batch melting formulation, and provide some sensitivity analysis to the chosen 1% factor. For reproducibility, I think a more detailed explanation of the melt-transport scheme is also required.

Good point, we do not use a melt transport model (now clarified on line 248), and so we do not resolve dynamic effects of unloading on the fraction of melt retained in the crystalline matrix. We have expanded on our choice of the retained melt fraction in the text (lines 431–37; 901–02; Figure S13). We also note that we do not resolve the rate of CO₂ release – only the magnitude of this effect due to one glacial cycle (clarified on lines 248–49). These are excellent topics for future research.

The combination of the above two concerns leads to my third concern, which is that the present deglaciation is affecting mostly the same mantle rock that was beneath Yellowstone for the previous deglaciation, because that mantle is upwelling rather slowly compared to Iceland.

We are not convinced that the rate at which mantle is being replaced beneath Yellowstone is small, under the assumption that Yellowstone is fed by a mantle plume. We expand on this in the new Section S5.

So where is the excess CO₂ sourced from? The bottom of the deepening melting regime, as hypothesised by Burley & Katz? Or simply by accelerating the segregation of the melt that was already in the melting region? Cerpa et al concluded that for MORs, the latter is much more important than the former. But this depends on the model for melt segregation, and also on the melting history of the solid mantle beneath Iceland. The authors seem to assume in their calculations that the mantle had previously undergone only steady upwelling, whereas they discuss the penultimate deglaciation being relevant for Yellowstone. I think a model of oscillatory glacial load would be a very nice addition, though this might be too much to ask. In any case, the authors should give some discussion and scaling analysis of the source of the excess CO₂ and the sensitivity of this to their assumptions. A corollary concern is that the mantle, having been depleted by the previous melting event during the previous deglacial, is now more refractory than assumed.

Because we do not resolve dynamic melt migration, the additional CO₂ is sourced from increased melting at depth (see Section S5; Figure S14).

The authors do provide comparison between the excess CO₂ delivered in their models with previous estimates. The rough agreement is taken as an implied validation of the models, but this may be misleading. Estimates of CO₂ degassed during the last deglacial may be dependent on the same assumptions made here.

We hope that the new sensitivity analyses (lines 289–302; Figure 5; Section S3.4; Figure S9 & S10) and further details on our method (lines 431–37) will clarify the strengths/limitations of our estimates, and justify/qualify the assumptions we make.

Other minor comments:

- Comparison of figures 2 and 4. Why are there no velocity vectors in the asthenosphere of figure 4?

The deeper velocity vectors are large and do not vary during deglaciation, so we plot only the shallow velocities at the relevant melting depths for visualization purposes. Clarified on lines 187–88.

Why is the decompression colour scale the same as in figure 2?

To easily compare the two figures.

Why are the red arrows in the legend of panels (a) so small? Does this represent the scale of the flow? If so, it makes the figure more difficult to understand.

Yes, the scale of the arrows corresponds to the magnitude, clarified on line 132. We have also made the arrows slightly bigger.

- Figure 3. Why is the total CO₂ rate perturbation larger for Yellowstone than for Iceland, and also greater over a longer period of time (2ka versus 1ka) and yet the estimates for the total CO₂ excess is 135--230 Gt in both cases (in the text)? These settings are very different, so shouldn't we expect rather different outcomes of deglaciation? If not, this would seem to call for a careful explanation, which seems to be missing.

The predicted CO₂ excess for Iceland is 0.30–1.21 Gt/km (roughly 30–121 Gt), and for Yellowstone is 18–79 Gt. The 135–230 Gt was our estimate for Yellowstone in the original manuscript, which is now smaller as we have used a smaller background flux (from Rahilly & Fischer, 2021), and smaller load radius. We think it makes sense that the CO₂ excess for Iceland is higher given the larger glacial load, more extensive deglacial melting, and the (now) similar background CO₂ fluxes.

- Lines 152-158 need references.

Added on lines 169 & 171.

- I found the discussion of dimensionality of models and results confusing. The models seem to be 2D Cartesian, which is a bit awkward for a plume model. But the authors seem to make some sort of 3D correction. I found all of this unclear.

In response to this comment and that of the other two reviewers, we now report the results in both 2-D and 3-D. We also further explore 3-D effects in Section S3.5, Figure S11.

- What is the definition of the "melt fraction rate"? I could understand the degree-of-melting

rate, because this is a property of the solid phase. Melt fraction is, in my mind, equal to porosity, which is a property of the two-phase system and therefore affected by both melting and segregation. So I don't see how it can be used as a proxy for melting alone.

Yes, the “melt fraction rate” is the “degree-of-melting rate”, we have clarified the definition on lines 87–88. In our study, the “melt fraction” is only a property of the solid phase and is not equivalent to the porosity.

- I liked the energetic comparisons that lead to the idea that magmatic heat transfer could contribute to ice melting and acceleration, but if this relies on excess melting due to deglaciation then the argument might be a bit circular...

Yes, the deglaciation is presumably driven and initiated by external climate factors (e.g., air temperatures), clarified on lines 275–77.

Reviewer #3 (Remarks to the Author):

This is a timely, well-designed, and well-executed modeling study on the links between glacial unloading, volcanism, and climate. Although this topic has been explored by a fair number of previous studies, this paper is novel in that it focuses on a comparison between two contrasting volcanic systems and their differing responses to ice unloading. The comparative approach to the modeling design allows for insights into dynamic processes that would not have been possible in a study of a single glaciovolcanic system. I think this will be an impactful paper that will be of wide interest to volcanologists, glacial geologists, climate scientists, and many other communities.

The paper is very well-written and thorough, and I have relatively few comments or suggestions for improvement. However, I offer some questions and thoughts on various details keyed to line numbers that are intended to add some clarity to the text. In my view, only minor edits are necessary before this is ready for publication.

Lines 60-62: Although the available evidence from the glacial and volcanic records does indicate the broad correspondence between glaciation and volcanism in Yellowstone, it may be important to point out that age control on many of the rhyolites and other volcanic units was developed some time ago (e.g., with K-Ar dating). Uncertainties on the ages of many of these units, along with uncertainties in the ages of the Bull Lake glacial deposits, makes it difficult to establish a precise timeline for glacio-volcanic processes during Bull Lake time. Limitations in age control also make it difficult to assess changes in rates of volcanic activity. Tighter age control is currently being developed for the volcanic units, although much of that information remains unpublished. Nonetheless, some of the current limitations should probably be noted here.

Good point, we note this on line 67–68.

Lines 79-81: I see that the geometry of the ice sheet is parameterized as a circular parabolic slab of ice thickness with a decreasing radius. This is a reasonable approximation for the Icelandic ice sheet, but not necessarily for the Yellowstone ice complex which consisted of several coalesced ice masses. I understand it may not be possible to incorporate this level of

detail in the modeling. I also see in the Methods section that various deglaciation styles were tested in the modeling. If space allows, I think it would be helpful if the authors can comment briefly here in the main text on whether the ice geometry simplifications they make would impart some level of uncertainty to the model outcomes they have generated. From the Methods section, I gather the answer is no.

This is a good point, and motivated us to run models in 3-D, assuming an elliptical ice cap geometry. We decided to adopt the short axis (50 km) of the ellipse in our 2-D (Cartesian) models, under the assumption that the ice retreat along the out-of-plane long axis is less important than the in-plane retreat (explained lines 89–90; 374–78). We test this assumption in Section S3.5, in which we compare the 2-D and 3-D models. We find that the 3-D effects are somewhat important (results in fluxes 70–120% compared to a circular load), but to delve more deeply into this would require improved/more detailed models of the Yellowstone deglaciation (e.g., Anderson et al., 2014) subject to age constraints. We also note that the timing of the retreat does not influence our estimate of cumulative additional melt produced (Figure S8). We have now examined the influence of ice radius in Figure 5; Figure S9b; Figure S10b.

Lines 247-248: I think it's important here to acknowledge and compare the relative influences of enhanced geothermal heat fluxes on ice flow and melting to the influences of warming climate on these same processes. The timing of deglaciation in both Yellowstone and Iceland is closely tied to rising atmospheric temperatures, implying that climatic controls play a dominant role in the deglaciation.

We agree and have clarified this point on line 275–767

Lines 261-263: It may also be relevant to point out that large hydrothermal explosions in Yellowstone occurred throughout the Holocene, suggesting triggers unrelated to ice unloading.

Yes, we add this point on line 283–84.

Line 266: I appreciate that the authors would like to place their findings in a broader context, and this concluding section accomplishes that goal. However, there is a fair amount of speculation here because there are no actual model results from West Antarctica and Greenland. It's fine to speculate, and many of the inferences and suggestions here are reasonable and follow from the modeling work described above. But it might be advisable to be clearer that additional modeling work will be necessary to support these broader inferences for WAIS and Greenland.

Yes, we add this point on line 335 (and back up this point using the new Figure 5b and lines 299–301).

References

- Anderson, L. S., Wickert, A. D., Colgan, W. T., & Anderson, R. S. (2014). Numerical Modeling of the Last Glacial Maximum Yellowstone Ice Cap Captures Asymmetry in Moraine Ages. In *2014 AGU Fall Meeting*. AGU.
- Eksinchol, I., Rudge, J. F., & Maclennan, J. (2019). Rate of Melt Ascent Beneath Iceland From the Magmatic Response to Deglaciation. *Geochemistry, Geophysics, Geosystems*. <https://doi.org/10.1029/2019GC008222>
- Haskell, N. A. (1935). The motion of a viscous fluid under a surface load. *Physics*, 6(8), 265–269.
- Ito, G., Shen, Y., Hirth, G., & Wolfe, C. J. (1999). Mantle flow, melting, and dehydration of the Iceland mantle plume. *Earth and Planetary Science Letters*, 165(1), 81–96.
- Jenkins, J., Maclennan, J., Green, R. G., Cottaar, S., Deuss, A. F., & White, R. S. (2018). Crustal formation on a spreading ridge above a mantle plume: receiver function imaging of the Icelandic crust. *Journal of Geophysical Research: Solid Earth*, 123(6), 5190–5208.
- Jull, M., & McKenzie, D. (1996). The effect of deglaciation on mantle melting beneath Iceland. *Journal of Geophysical Research: Solid Earth*.
- Rahilly, K. E., & Fischer, T. P. (2021). Total diffuse CO₂ flux from Yellowstone caldera incorporating high CO₂ emissions from cold degassing sites. *Journal of Volcanology and Geothermal Research*, 419, 107383. <https://doi.org/https://doi.org/10.1016/j.jvolgeores.2021.107383>
- Werner, C., & Brantley, S. (2003). CO₂ emissions from the Yellowstone volcanic system. *Geochemistry, Geophysics, Geosystems*. <https://doi.org/10.1029/2002GC000473>

Reviewer #1 (Remarks to the Author):

The authors have presented an improved manuscript in response to the reviewers' comments and the logic of the work is now much easier to follow. I appreciate the explanation of the modelling approach and the presentation of sensitivity testing to lithospheric rheology.

The key conclusion remains that the model results indicate that Yellowstone may have seen important variations in the magmatic CO₂ flux associated with glacial cycles – with melt generated during glacial unloading carrying 20-80 Gt CO₂ in that setting. The missing piece of the argument is the lack of observational evidence for either an increase in volcanic productivity or increased magmatic CO₂ flux at Yellowstone at that time. Perhaps one of the goals of the study is to encourage observational work to explore the signal. This is risky, however, as the authors note that the thick lithosphere and established magmatic plumbing system in continental crust – including large shallow rhyolitic chambers – may act to filter any important input from the mantle.

I remain confused by the approach to estimating melt and CO₂ fluxes in Iceland. They use Iceland to validate their model, so it is important that link between model results and available observations is properly understood.

Magma Fluxes: In terms of magma fluxes, I am still troubled by the way that they switch between 2D and 3D models and observations for Iceland. In the abstract, the main text and supplementary info they clearly focus their efforts on a comparison of the models with Iceland as a whole (not some limited portion of it that they think might sit above the plume conduit). On line 98 of the updated manuscript they report a 2D steady-state melt flux of 2.3×10^{-3} km²/yr – equivalent to 0.23 km³/yr volumetric melt flux over a 100 km long ridge segment. This calculation is revisited in lines 379-392 of the revised manuscript in the section labelled "Iceland model" and their response to point 1 of my original review.

I don't agree with the arguments in their response. They argue that the huge melt flux associated with the plume is redistributed along-axis (out of the 2D plane of the model) –

"We chose to multiply by a 100-km along axis length-scale for Iceland, as this is the plume diameter over which the temperature structure stays relatively constant in 3-D models (e.g., Ito et al., 1999)."

Maybe – it depends what depth is chosen for the model slice and also which model is used – in Ito's 2001 Nature paper he has potential temperatures that vary by less than about 50 degrees between the plume centre and a position >250 km down the ridge axis. Jenkins et al (EPSL, 2016) show that the 410 km discontinuity depth anomaly occupies a similar area to Iceland. So I am not seeing the fact that Ito fixed a ~100 radius plume conduit as a clear justification for the authors' approach to conversion from a 2D model to 3D melt flux.

"We do not consider contributions further away from the plume head. We do agree that the melt produced in our model cannot be emplaced within this 100-km zone nearest to the plume head, as this would imply crustal thicknesses that are too large. We expect this extra material is emplaced laterally along the rift zone (see lines 400-01)."

While there is some minor redistribution of melt along the ridge axes, this is not likely to be a volumetrically important process on a scale of many tens of kilometres along-axis. The geochemistry and geology of Iceland provide observational constraints on the importance of and length-scale of along-axis melt redistribution. MacLennan et al 2001 (EPSL – plume driven upwelling under central Iceland) demonstrated that the observed trace element geochemistry of the basalts and crustal thickness variations could be matched without redistribution along the northern volcanic zone. Later geochemical

work has confirmed that each volcanic system has a distinctive Pb-isotope geochemistry that is consistent with minimal along-axis melt redistribution (e.g. Shorttle et al., GCA, 2013). So, this along-axis redistribution cannot be called upon to resolve the flux mismatch in their models. The Ito paper referred to by the authors as evidence for melt redistribution predates these observationally-based articles.

“If most of the material is produced within 100-km of the plume head and emplaced within 300 km, the 128-km crustal thickness would be ~40 km, consistent with seismic observations (e.g., Jenkins et al., 2018). We clarify this on line 403–405.”

I find this argument confusing. First, the average crustal thickness on the rift zones from the Jenkins paper onland Iceland is about 30 km. The thickness of 40 km that is mentioned is the maximum in central Iceland. Second, the suggestion here is that somehow that no mantle melting takes place beneath the spreading centre at radii between 100 and 300 from the plume centre. This seems very unlikely to me – passive (plate-driven) upwelling of mantle with a T_p of about 1500 deg C generates about 20 km of crust – very similar to that observed at the coast of Iceland.

After all this, the worry for me is that their steady-state model for Iceland does not appear to do a good job of matching the overall melt flux from the island of Iceland. All of the exposed rift-zones of Iceland were glaciated and they whole length of the zone of plate separation needs to be included in the calculations somehow. This is important, because their model is aimed at matching Icelandic observations as a validation exercise – if they cannot find a way of matching the Icelandic observations this throws some doubt onto their conclusions.

CO₂ contents and fluxes: The CO₂ flux estimate for Iceland comes from Barry et al (GCA 2014) and provide fluxes with huge uncertainties 0.2-2.3x10¹⁰ mol/yr for all of the rift zones of Iceland. Once again, it does not make sense to divide by 100 km here because the flux corresponds to the island as a whole. The huge uncertainties provided by the Barry estimate make the comparison of model results and observations less useful than the melt flux one might be as part of a model validation exercise.

Figure S12a – the data labels are not correct and have been switched for subglacial and post-glacial data. The subglacial samples have higher REE concentrations than the (early) postglacial ones.

Reviewer #2 (Remarks to the Author):

The manuscript is substantially improved by the authors' thorough and careful consideration of reviewer comments. The changes made in response to my own comments address them more than adequately. I am very happy to endorse the manuscript for publication at this stage.

However, I would suggest that the authors take the opportunity to reconsider their use of "melt fraction" when they mean "degree of melting." These terms have well-established meaning in the literature (despite some unfortunate deviations). Because the two concepts differ in an important but subtle manner, I think it is crucial to use clear language for this. The authors will make their paper more reader-friendly by sticking with standard usage.

Reviewer #3 (Remarks to the Author):

I appreciate the opportunity to evaluate the revised version of this intriguing manuscript by Clerc, Behn, and Minchew. After having read through all the responses to my original

review and the revised manuscript, I believe the authors have properly addressed and resolved all of my questions and critiques (which were quite minor) on the first version.

I have just a few small additional questions and suggestions listed below, mainly concerning paper citations. These should be straightforward to resolve.

I welcome the improvements and clarifications in this revision, and I support publication of the manuscript after very minor changes.

Joe Licciardi

Enumerated comments:

In Figure 3 panel a, why are ice volume units expressed as km²? I see comments in the supplement that distinguish "ice volumes in 2-D" and "radially-integrated ice volumes," but unless I missed it, this distinction is not clearly explained in the main text and is somewhat confusing.

Lines 275-277: The Young et al. (2011) paper provides a good overview of late Pleistocene climate forcings related to deglaciation across the western US. But for climate-driven deglaciation of the Yellowstone glacial system specifically, I suggest citing the more recent Licciardi & Pierce (2018) paper that summarizes the geochronological evidence for Yellowstone's glacial history and climatic influences on glacier retreat.

Lines 283-284: Regarding the timing of hydrothermal explosions, Pierce et al (2002) includes a valuable compilation and discussion of age control for many of the explosion deposits, but I think it is also important to cite the following (and more recent) report on this topic:

Morgan, L.A., Shanks III, W.C., Pierce, K.L., 2009. Hydrothermal processes above the Yellowstone magma chamber: large hydrothermal systems and large hydrothermal explosions. Geol. Soc. Am. Spec. Pap. 459, 1-95.

Line 603: Incorrect reference information here. Correct info is:

Pierce, K.L., Cannon, K.P., Meyer, G.A., Trebesch, M.J., Watts, R.D., 2002. Post-Glacial Inflation-Deflation Cycles, Tilting, and Faulting in the Yellowstone Caldera Based on Yellowstone Lake Shorelines. U.S. Geological Survey Open-File Report 02-0142.

Reviewer 1 has raised reasonable concerns with the manuscript that address a level of detail that my own review did not. Although I don't disagree with these concerns and I feel that better addressing them could strengthen the manuscript, I nonetheless recommend accepting the manuscript for publication. In my view, the work under review makes a reasonable first approximation of the possible CO₂ emissions from Yellowstone during deglaciation. The idea itself deserves wide attention, and the models are plausible if imperfect. More details are interspersed below.

Reviewer #1:

The key conclusion remains that the model results indicate that Yellowstone may have seen important variations in the magmatic CO₂ flux associated with glacial cycles – with melt generated during glacial unloading carrying 20-80 Gt CO₂ in that setting. The missing piece of the argument is the lack of observational evidence for either an increase in volcanic productivity or increased magmatic CO₂ flux at Yellowstone at that time. Perhaps one of the goals of the study is to encourage observational work to explore the signal. This is risky, however, as the authors note that the thick lithosphere and established magmatic plumbing system in continental crust – including large shallow rhyolitic chambers – may act to filter any important input from the mantle.

Unclear what the reviewer means by “risky” here. Is it risky to publish an important idea even if a direct observational test of the idea is currently unavailable or difficult? I don't see how that is risky, except in that the idea might gain traction before it is supported by observations.

I remain confused by the approach to estimating melt and CO₂ fluxes in Iceland. They use Iceland to validate their model, so it is important that link between model results and available observations is properly understood.

I think a weakness of the modelling is that it makes no attempt to track melt transport. Melt extraction is assumed to be instantaneous and without pathway — so everything is determined by the melting rate. This isn't as good as some of the work out of the Cambridge group, but it is an appropriate first attempt, consistent with the Jull and McKenzie approach that started this all off.

Magma Fluxes: In terms of magma fluxes, I am still troubled by the way that they switch between 2D and 3D models and observations for Iceland. In the abstract, the main text and supplementary info they clearly focus their efforts on a comparison of the models with Iceland as a whole (not some limited portion of it that they think might sit above the plume conduit). On line 98 of the updated manuscript they report a 2D steady-state melt flux of 2.3×10^{-3} km²/yr – equivalent to 0.23 km³/yr volumetric melt flux over a 100 km long ridge segment. This calculation is revisited in lines 379-392 of the revised manuscript in the section labelled “Iceland model” and their response to point 1 of my original review.

I don't agree with the arguments in their response. They argue that the huge melt flux associated with the plume is redistributed along-axis (out of the 2D plane of the model) –

“We chose to multiply by a 100-km along axis length-scale for Iceland, as this is the plume diameter over which the temperature structure stays relatively constant in 3-D models (e.g., Ito et al., 1999).”

This is indeed a weakness of the paper, as noted above. But it isn't clear that all melt produced should be erupted. I think the melt redistribution idea was added in response to the reviewer comments from the first round, and the authors might have done a more convincing job of it.

Maybe – it depends what depth is chosen for the model slice and also which model is used – in Ito's 2001 Nature paper he has potential temperatures that vary by less than about 50 degrees between the plume centre and a position >250 km down the ridge axis. Jenkins et al (EPSL, 2016) show that the 410 km discontinuity depth anomaly occupies a similar area to Iceland. So I am not seeing the fact that Ito fixed a ~100 radius plume conduit as a clear justification for the authors' approach to conversion from a 2D model to 3D melt flux.

Well, I think that the authors have made reasonable choices but could have made other reasonable choices. The reviewer is expecting too much here.

"We do not consider contributions further away from the plume head. We do agree that the melt produced in our model cannot be emplaced within this 100-km zone nearest to the plume head, as this would imply crustal thicknesses that are too large. We expect this extra material is emplaced laterally along the rift zone (see lines 400–01)."

While there is some minor redistribution of melt along the ridge axes, this is not likely to be a volumetrically important process on a scale of many tens of kilometres along-axis. The geochemistry and geology of Iceland provide observational constraints on the importance of and length-scale of along-axis melt redistribution. Maclennan et al 2001 (EPSL – plume driven upwelling under central Iceland) demonstrated that the observed trace element geochemistry of the basalts and crustal thickness variations could be matched without redistribution along the northern volcanic zone. Later geochemical work has confirmed that each volcanic system has a distinctive Pb-isotope geochemistry that is consistent with minimal along-axis melt redistribution (e.g. Shorttle et al., GCA, 2013). So, this along-axis redistribution cannot be called upon to resolve the flux mismatch in their models. The Ito paper referred to by the authors as evidence for melt redistribution predates these observationally-based articles.

Reasonable arguments but I think they are not as watertight as the reviewer implies. Leeway is appropriate here, I think.

"If most of the material is produced within 100-km of the plume head and emplaced within 300 km, the 128-km crustal thickness would be ~40 km, consistent with seismic observations (e.g., Jenkins et al., 2018). We clarify this on line 403–405."

I find this argument confusing. First, the average crustal thickness on the rift zones from the Jenkins paper onland Iceland is about 30 km. The thickness of 40 km that is mentioned is the maximum in central Iceland. Second, the suggestion here is that somehow that no mantle melting takes place beneath the spreading centre at radii between 100 and 300 from the plume centre. This seems very unlikely to me – passive (plate-driven) upwelling of mantle with a T_p of about 1500 deg C generates about 20 km of crust – very similar to that observed at the coast of Iceland.

Here I think the reviewer is correct that the authors haven't been clear about consistency with the literature, and should revise their wording to be more open about the discrepancy and associated issues.

After all this, the worry for me is that their steady-state model for Iceland does not appear to do a good job of matching the overall melt flux from the island of Iceland. All of the exposed rift-zones of Iceland were glaciated and they whole length of the zone of plate separation needs to be included in the calculations somehow. This is important, because their model is aimed at matching Icelandic observations as a validation exercise – if they cannot find a way of matching the Icelandic observations this throws some doubt onto their conclusions.

OK, this is true, but again, I think a discussion of the discrepancy and its causes is the way forward, not a rejection of the ideas.

CO2 contents and fluxes: The CO2 flux estimate for Iceland comes from Barry et al (GCA 2014) and provide fluxes with huge uncertainties $0.2-2.3 \times 10^{10}$ mol/yr for all of the rift zones of Iceland. Once again, it does not make sense to divide by 100 km here because the flux corresponds to the island as a whole. The huge uncertainties provided by the Barry estimate make the comparison of model results and observations less useful than the melt flux one might be as part of a model validation exercise.

I agree that the redistribution along axis is a bit awkward — but I think it was done in response to the reviews.

My recommendation:

1) Another round of revision

2) The authors are clearer about the uncertainties in their models and the discrepancies with observations

3) The authors give more weight and discussion to the excellent work by the Cambridge group (Maclennan, Shorttle, Rudge and others) on the Iceland data and models.

We thank the reviewers for their comments, which we address point-by-point (in blue) below. Line numbers refer to those in the “tracked changes” version of the revised manuscript.

REVIEWER COMMENTS

Reviewer #1 (Remarks to the Author):

The authors have presented an improved manuscript in response to the reviewers' comments and the logic of the work is now much easier to follow. I appreciate the explanation of the modelling approach and the presentation of sensitivity testing to lithospheric rheology.

The key conclusion remains that the model results indicate that Yellowstone may have seen important variations in the magmatic CO₂ flux associated with glacial cycles – with melt generated during glacial unloading carrying 20-80 Gt CO₂ in that setting. The missing piece of the argument is the lack of observational evidence for either an increase in volcanic productivity or increased magmatic CO₂ flux at Yellowstone at that time. Perhaps one of the goals of the study is to encourage observational work to explore the signal. This is risky, however, as the authors note that the thick lithosphere and established magmatic plumbing system in continental crust – including large shallow rhyolitic chambers – may act to filter any important input from the mantle.

We agree with the need for future observational work (which we hope this work may further motivate), and as noted by Reviewer #1. The manuscript clearly states the lack of observational evidence, including in the abstract (lines 16–18; lines 332–339).

To further highlight the uncertainties in the estimate of enhanced melt volumes beneath Yellowstone, we now report a range of uncertainties of 250 – 620 km³ (lines 23 and 284) associated with model geometry and background melt production rates. We have added text on lines 282–284 to acknowledge this and to point the reader to the sensitivity analyses in the supplement. We more explicitly retitle sections of the supplement as “sensitivity analyses” (lines 995 & 1064). We also recognize that our reported value of 364 km³ of extra melt generated during the Yellowstone deglaciation implies a much greater degree of precision than is possible given parameter uncertainties, so we instead report 360 km³ (lines 281 & 353).

I remain confused by the approach to estimating melt and CO₂ fluxes in Iceland. They use Iceland to validate their model, so it is important that link between model results and available observations is properly understood.

Magma Fluxes: In terms of magma fluxes, I am still troubled by the way that they switch between 2D and 3D models and observations for Iceland. In the abstract, the main text and supplementary info they clearly focus their efforts on a comparison of the models with Iceland as a whole (not some limited portion of it that they think might sit above the plume conduit). On line 98 of the updated manuscript they report a 2D steady-state melt flux of 2.3×10^{-3} km²/yr – equivalent to 0.23 km³/yr volumetric melt flux over a 100 km

long ridge segment. This calculation is revisited in lines 379-392 of the revised manuscript in the section labelled "Iceland model" and their response to point 1 of my original review.

I don't agree with the arguments in their response. They argue that the huge melt flux associated with the plume is redistributed along-axis (out of the 2D plane of the model) "We chose to multiply by a 100-km along axis length-scale for Iceland, as this is the plume diameter over which the temperature structure stays relatively constant in 3-D models (e.g., Ito et al., 1999)."

Maybe – it depends what depth is chosen for the model slice and also which model is used – in Ito's 2001 Nature paper he has potential temperatures that vary by less than about 50 degrees between the plume centre and a position >250 km down the ridge axis. Jenkins et al (EPSL, 2016) show that the 410 km discontinuity depth anomaly occupies a similar area to Iceland. So I am not seeing the fact that Ito fixed a ~100 radius plume conduit as a clear justification for the authors' approach to conversion from a 2D model to 3D melt flux.

"We do not consider contributions further away from the plume head. We do agree that the melt produced in our model cannot be emplaced within this 100-km zone nearest to the plume head, as this would imply crustal thicknesses that are too large. We expect this extra material is emplaced laterally along the rift zone (see lines 400–01)."

While there is some minor redistribution of melt along the ridge axes, this is not likely to be a volumetrically important process on a scale of many tens of kilometres along-axis. The geochemistry and geology of Iceland provide observational constraints on the importance of and length-scale of along-axis melt redistribution. Maclennan et al 2001 (EPSL – plume driven upwelling under central Iceland) demonstrated that the observed trace element geochemistry of the basalts and crustal thickness variations could be matched without redistribution along the northern volcanic zone. Later geochemical work has confirmed that each volcanic system has a distinctive Pb-isotope geochemistry that is consistent with minimal along-axis melt redistribution (e.g. Shorttle et al., GCA, 2013). So, this along-axis redistribution cannot be called upon to resolve the flux mismatch in their models. The Ito paper referred to by the authors as evidence for melt redistribution predates these observationally-based articles.

"If most of the material is produced within 100-km of the plume head and emplaced within 300 km, the 128-km crustal thickness would be ~40 km, consistent with seismic observations (e.g., Jenkins et al., 2018). We clarify this on line 403–405."

I find this argument confusing. First, the average crustal thickness on the rift zones from the Jenkins paper onland Iceland is about 30 km. The thickness of 40 km that is mentioned is the maximum in central Iceland. Second, the suggestion here is that somehow that no mantle melting takes place beneath the spreading centre at radii between 100 and 300 from the plume centre. This seems very unlikely to me – passive (plate-driven) upwelling of mantle with a T_p of about 1500 deg C generates about 20 km of crust – very similar to that observed at the coast of Iceland.

After all this, the worry for me is that their steady-state model for Iceland does not appear to do a good job of matching the overall melt flux from the island of Iceland. All of the exposed rift-zones of Iceland were glaciated and they whole length of the zone of plate separation needs to be included in the calculations somehow. This is important,

because their model is aimed at matching Icelandic observations as a validation exercise – if they cannot find a way of matching the Icelandic observations this throws some doubt onto their conclusions.

We thank Reviewer #1 for bringing up this issue, we address this concern by introducing a “pooling width”. In the revised manuscript, we resolve the discrepancy between our model results for melt production in Iceland and available observations of crustal thickness, by assuming only melt generated within 30 km of the ridge axis reaches the surface. We explain this modification in the methods (lines 551–75 and 641–47) and further justify and explore the effect of this pooling width in the supplemental information (Section S2.3, Figure S6). We have updated all relevant values of melt production, CO₂ flux, and heat release for Iceland (these are now smaller).

The concept of a maximum distance that melts can migrate to the ridge axis has been explored extensively in the literature (e.g., Hebert & Montési, 2010; Montési et al., 2011; Behn & Grove, 2015). These studies show that particularly in melt-rich environments, full pooling of all melts significantly over-predicts crustal thickness and produces a mismatch with the composition of pooled melts. The maximum pooling width of 30 km that we assume is consistent with estimates from Hebert & Montési, (2010) and Behn & Grove (2015). This lowers our estimate of crustal production in the Iceland model, and thus we have eliminated our earlier argument about along-axis melt redistribution. These modifications are explained in the manuscript on lines 551–75, lines 641–47, section S2.3, and Figure S6.

We do not modify the Yellowstone model, as our modeled melt generation is benchmarked against estimates of mantle melt generation (e.g., the melt generation required to match chloride and CO₂ flux observations), as opposed to observations of crustal thickness for Iceland. Nevertheless, we do perform a similar exploration of the effect of pooling width on melt generation/CO₂ release in Yellowstone in Section S3.6 and Figure S10. We also further clarify that our model for Iceland is intended to validate/place the Yellowstone model into context on lines 106–14.

CO₂ contents and fluxes: The CO₂ flux estimate for Iceland comes from Barry et al (GCA 2014) and provide fluxes with huge uncertainties 0.2-2.3x10¹⁰ mol/yr for all of the rift zones of Iceland. Once again, it does not make sense to divide by 100 km here because the flux corresponds to the island as a whole. The huge uncertainties provided by the Barry estimate make the comparison of model results and observations less useful than the melt flux one might be as part of a model validation exercise.

We agree that the 2–D to 3–D scaling makes the comparison between our modelled flux and that estimated by Barry et al. (2014) difficult. We have added text to acknowledge this on lines 282–84. We also remove the Barry et al. estimate from Figure 3. We further clarify that the scaling is only useful as an approximate comparison (lines 551–67).

Figure S12a – the data labels are not correct and have been switched for subglacial and post-glacial data. The subglacial samples have higher REE concentrations than the (early) postglacial ones.

These data are reported correctly, as in the original Maclennan et al. (2002) publication. The “postglacial” data is more depleted as it refers to lavas aged 12.5 – 10 ka (shortly following the deglaciation), whereas the “subglacial” data refers to lavas aged 74 – 12.5 ka. This is now clarified in the figure caption (line 1352).

Reviewer #2 (Remarks to the Author):

The manuscript is substantially improved by the authors' thorough and careful consideration of reviewer comments. The changes made in response to my own comments address them more than adequately. I am very happy to endorse the manuscript for publication at this stage.

However, I would suggest that the authors take the opportunity to reconsider their use of "melt fraction" when they mean "degree of melting." These terms have well-established meaning in the literature (despite some unfortunate deviations). Because the two concepts differ in an important but subtle manner, I think it is crucial to use clear language for this. The authors will make their paper more reader-friendly by sticking with standard usage.

We now further clarify this in the Figure 1 color bar and caption (line 143).

Reviewer #3 (Remarks to the Author):

I appreciate the opportunity to evaluate the revised version of this intriguing manuscript by Clerc, Behn, and Minchew. After having read through all the responses to my original review and the revised manuscript, I believe the authors have properly addressed and resolved all of my questions and critiques (which were quite minor) on the first version.

I have just a few small additional questions and suggestions listed below, mainly concerning paper citations. These should be straightforward to resolve.

I welcome the improvements and clarifications in this revision, and I support publication of the manuscript after very minor changes.

Joe Licciardi

Enumerated comments:

In Figure 3 panel a, why are ice volume units expressed as km²? I see comments in the supplement that distinguish "ice volumes in 2-D" and "radially-integrated ice volumes,"

but unless I missed it, this distinction is not clearly explained in the main text and is somewhat confusing.

The ice volumes are in 2–D as the models shown in Figure 3 are 2–D (there is no 3–D Iceland model, and we worried it may be confusing to show 2–D volumes for Iceland next to equivalent 3–D volumes for Yellowstone). In the Figure 3 caption (line 259), we now refer the reader to Figure S9b, in which the radially-integrated 3-D ice volumes for Yellowstone are shown.

Lines 275-277: The Young et al. (2011) paper provides a good overview of late Pleistocene climate forcings related to deglaciation across the western US. But for climate-driven deglaciation of the Yellowstone glacial system specifically, I suggest citing the more recent Licciardi & Pierce (2018) paper that summarizes the geochronological evidence for Yellowstone's glacial history and climatic influences on glacier retreat.

Citation changed.

Lines 283-284: Regarding the timing of hydrothermal explosions, Pierce et al (2002) includes a valuable compilation and discussion of age control for many of the explosion deposits, but I think it is also important to cite the following (and more recent) report on this topic:

Morgan, L.A., Shanks III, W.C., Pierce, K.L., 2009. Hydrothermal processes above the Yellowstone magma chamber: large hydrothermal systems and large hydrothermal explosions. Geol. Soc. Am. Spec. Pap. 459, 1-95.

Citation added.

Line 603: Incorrect reference information here. Correct info is:

Pierce, K.L., Cannon, K.P., Meyer, G.A., Trebesch, M.J., Watts, R.D., 2002. Post-Glacial Inflation-Deflation Cycles, Tilting, and Faulting in the Yellowstone Caldera Based on Yellowstone Lake Shorelines. U.S. Geological Survey Open-File Report 02-0142.

Thank you, corrected.

References

Barry, P. H., Hilton, D. R., Furi, E., Halldórsson, S. A. & Grönvold, K. Carbon isotope and abundance systematics of Icelandic geothermal gases, fluids and subglacial basalts with implications for mantle plume-related CO₂ fluxes. *Geochim. Cosmochim. Acta* (2014). doi:10.1016/j.gca.2014.02.038

Behn, M. D. & Grove, T. L. Melting systematics in mid-ocean ridge basalts: Application of a plagioclase-spinel melting model to global variations in major element chemistry and crustal thickness. *J. Geophys. Res. Solid Earth* 120, 4863–4886 (2015).

Hebert, L. B. & Montési, L. G. J. Generation of permeability barriers during melt extraction at mid-ocean ridges. *Geochemistry, Geophys. Geosystems* 11, (2010).

Montési, L. G. J., Behn, M. D., Hebert, L. B., Lin, J. & Barry, J. L. Controls on melt migration and extraction at the ultraslow Southwest Indian Ridge 10–16 E. *J. Geophys. Res. Solid Earth* 116, (2011).